# *APOE4* is associated with elevated blood lipids and lower levels of innate immune biomarkers in a tropical Amerindian subsistence population

Angela R Garcia[1,2]*, Caleb Finch[3], Margaret Gatz[4], Thomas Kraft[5], Daniel Eid Rodriguez[6], Daniel Cummings[7], Mia Charifson[8], Kenneth Buetow[1,9], Bret A Beheim[10], Hooman Allayee[11,12], Gregory S Thomas[13], Jonathan Stieglitz[14], Michael D Gurven[5†], Hillard Kaplan[7†], Benjamin C Trumble[15]*†

[1]Center for Evolution and Medicine, Arizona State University, Tempe, United States; [2]Department of Anthropology, Emory University, Atlanta, United States; [3]Leonard Davis School of Gerontology, Dornsife College, University of Southern California, Los Angeles, Los Angeles, United States; [4]Center for Economic and Social Research, University of Southern California, Los Angeles, Los Angeles, United States; [5]Department of Anthropology, University of California, Santa Barbara, Santa Barbara, United States; [6]Department of Medicine, Universidad de San Simón, Cochabamba, Bolivia; [7]Institute for Economics and Society, Chapman University, Orange, United States; [8]Vilcek Institute of Graduate Biomedical Sciences, New York University, New York, United States; [9]School of Life Sciences, Arizona State University, Tempe, United States; [10]Department of Human Behavior, Ecology and Culture, Max Planck Institute for Evolutionary Anthropology, Leipzig, Germany; [11]Department of Preventive Medicine and Biochemistry & Molecular Medicine, Keck School of Medicine, University of Southern California, Irvine, Irvine, United States; [12]Department of Preventive Medicine, Keck School of Medicine, University of Southern California, Irvine, Irvine, United States; [13]Long Beach Memorial, Long Beach and University of California Irvine, Irvine, United States; [14]Institute for Advanced Study in Toulouse, Universite Toulouse, Toulouse, France; [15]School of Human Evolution and Social Change, Arizona State University, Tempe, United States

*For correspondence:
Angela.Garcia.2@asu.edu (ARG);
trumble@asu.edu (BCT)

†These authors contributed equally to this work

**Abstract** In post-industrial settings, apolipoprotein *E4* (*APOE4*) is associated with increased cardiovascular and neurological disease risk. However, the majority of human evolutionary history occurred in environments with higher pathogenic diversity and low cardiovascular risk. We hypothesize that in high-pathogen and energy-limited contexts, the *APOE4* allele confers benefits by reducing innate inflammation when uninfected, while maintaining higher lipid levels that buffer costs of immune activation during infection. Among Tsimane forager-farmers of Bolivia (*N* = 1266, 50 % female), *APOE4* is associated with 30 % lower C-reactive protein, and higher total cholesterol and oxidized LDL. Blood lipids were either not associated, or negatively associated with inflammatory biomarkers, except for associations of oxidized LDL and inflammation which were limited to obese adults. Further, *APOE4* carriers maintain higher levels of total and LDL cholesterol at low body mass indices (BMIs). These results suggest that the relationship between *APOE4* and lipids may be beneficial for pathogen-driven immune responses and unlikely to increase cardiovascular risk in an active subsistence population.

**eLife digest** Genes contain the instructions needed for a cell to make molecules called proteins, which perform various roles in the body. Different variants of a gene can affect how the protein works, and in some cases, can increase a person's risk to develop certain diseases.

For example, people who carry a version of the apolipoprotein E gene called APOE4 have a greater risk of developing Alzheimer's disease or heart disease. Individuals with two copies of this genetic variant have a 45% higher risk of heart disease and 12 times higher risk of Alzheimer's disease. Studies in industrialized countries suggest this increased risk may be the result of higher cholesterol and inflammation in people with APOE4. But if APOE4 is harmful, why does it continue to be so common worldwide?

One potential explanation is that APOE4, which has been around since before modern humans, may be beneficial in some contexts. Cholesterol is essential for many vital tasks in the body. In physically demanding environments where parasitic infections are common – conditions similar to those experienced by early humans – APOE4 might be beneficial. Under those circumstances, having more cholesterol might help fuel metabolic activities, fight infections, or reduce inflammation caused by infections.

Garcia et al. investigated the link between the APOE4 genetic variant, cholesterol and inflammation in 1,266 Indigenous Tsimane people from 80 villages in Bolivia. Tsimane people live an active lifestyle foraging and farming for food. Parasite infections are a common problem in their communities, but obesity rates are very low. Garcia et al. found that Tsimane people with at least one copy of the APOE4 have lower levels of inflammation and higher levels of cholesterol than those who have two copies of the APOE3 version of the gene. Very lean people with APOE4 had especially high levels of the so called "bad" low density lipoprotein (LDL) cholesterol compared to people with APOE3 only. However, in this situation, storing a little extra cholesterol may not be so bad.

The findings contradict other studies that have linked obesity to higher LDL levels and APOE4 to higher levels of inflammation. For the majority of human history, humans lived in more physically strenuous and calorically restrictive environments, with less access to clean water. Garcia et al. suggest that the harmful effects of APOE4 seen in studies in more industrialized societies – where people tend to be more sedentary and have less exposure to pathogens – may reflect a mismatch between a person's environment and their genes. More studies that capture the diversity of environmental conditions under which people live will help clarify the role of APOE4 health and disease.

## Introduction

The apolipoprotein *E4* (*APOE4*) allele is considered a major shared risk factor for both cardiovascular disease (CVD) (*Hansson and Libby, 2006*) and Alzheimer's disease (AD) (*Belloy et al., 2019*; *Smith et al., 2019*), in part due to its role in lipid metabolism and related inflammation (*Huebbe and Rimbach, 2017*). *APOE4+* carriers consistently show higher levels of total cholesterol, low-density lipoprotein (LDL), and oxidized LDL (*Safieh et al., 2019*; *Yassine and Finch, 2020*). While some studies suggest *APOE4+* carriers have higher inflammatory responses (*Gale et al., 2014*; *Olgiati et al., 2010*), the *APOE4* allele is also associated with downregulation of aspects of innate immune function, including acute phase proteins (*Lumsden et al., 2020*; *Martiskainen et al., 2018*; *Vasunilashorn et al., 2011*), and toll-like receptor (TLR) signaling molecules (*Dose et al., 2018*).

*APOE*, lipids, and immune function may interact differently in contemporary obesogenic post-industrial contexts compared to environments where infections are prevalent. For instance, a prospective U.S.-based cohort study (Framingham Heart Study) found that individuals with *APOE4* and low-grade chronic obesity-related inflammation had higher risk of developing AD, with earlier onset than *APOE3/3* and *APOE4+* carriers without inflammation (*Tao et al., 2018*). By contrast, *APOE4* may protect against cognitive loss among those infected with parasites (*Oriá et al., 2005*; *Trumble et al., 2017*) and accelerate recovery from viral infection (*Mueller et al., 2016*).

These findings suggest that interactive influences of *APOE4,* lipids, and immune function on disease risks may be environmentally moderated. However, this is difficult to test because most biomedical research is conducted in controlled laboratory settings using animal models, or in post-industrial populations with low-pathogen burden and high obesity prevalence (*Gurven and Lieberman, 2020*). Here,

we begin to fill this gap by evaluating both immune and lipid profiles of individuals with *APOE3/3* and *APOE4+* genotypes living in a pathogenically diverse, energy-limited environment.

## *APOE*, cholesterols, and immune function

*APOE* has three functionally polymorphic allelic variants: *E4*, *E3*, and *E2* (*Demarchi et al., 2005*; *Safieh et al., 2019*). The most prevalent, *APOE3*, arose ~200 K years ago from a single nucleotide polymorphism (SNP) (C → T) at locus 19q13 from the ancestral *APOE4* (*Fullerton et al., 2000*). The evolutionary success of *APOE3* has been attributed to its greater plasticity in response to environmental changes compared to the ancestral *APOE4* allele (*Huebbe and Rimbach, 2017*). However, *APOE4* is maintained at frequencies up to 40%, such as those in some central African populations.

Maintenance and distribution of *APOE* variants may in part be due to the distinct functional capabilities of allelic variants (*Trotter et al., 2010*). Eisenberg and colleagues have suggested that the benefits of *APOE4* may be most appreciable in environments where cholesterol requirements are increased to meet higher metabolic demands, such as at high and low latitudes where there are climatic extremes (e.g. northern Europe and South America) (*Eisenberg et al., 2010*).

Another leading explanation for *APOE4* persistence is based on the theory of antagonistic pleiotropy (*Williams, 1957*), which posits that the *APOE4* allele may persist due to the fitness benefits of lipid buffering in early life relative to *APOE3*, outweighing any harmful health effects that manifest in a post-reproductive life stage (i.e. 'selection's shadow') (*Smith et al., 2019*). Consistent with the notion of early life fitness advantage, the *APOE4* variant is associated with lower infant mortality and higher fertility among rural Ghanaians experiencing high-pathogen burden (*van Exel et al., 2017*).

Innate immune responses with fever and systemic inflammation are also energetically expensive (*Muehlenbein et al., 2010*), and cholesterol and other lipids are necessary for fueling these responses (*Tall and Yvan-Charvet, 2014*). Helminths and some protozoal parasites also require lipids to fuel their own metabolic activities, either pulling them from the host bloodstream, or modulating host metabolic activities like LDL endocytosis or lipid remodeling to gain access to lipids (*Bansal et al., 2005*). Thus, in high-pathogen and energy-limited environments, where there may be persistent pathogen-driven immune activation, the ability to maintain peripheral cholesterol levels would presumably be a benefit throughout life (*Finch and Martin, 2016*; *Gurven et al., 2016*).

Despite *APOE4* being associated with neuroinflammation among those with AD (*Kloske and Wilcock, 2020*), in 'healthy' individuals the *APOE4* allele is associated with downregulated innate immunity. Specifically, blood levels of C-reactive protein (CRP) in *APOE4+* carriers are lower in several post-industrial populations (*Lumsden et al., 2020*; *Martiskainen et al., 2018*), as well as the Tsimane, an Amerindian population in rural Bolivia (and focus of the present study) (*Vasunilashorn et al., 2011*). Moreover, among the Tsimane, *APOE4+* carriers had lower levels of blood eosinophils (*Trumble et al., 2017*; *Vasunilashorn et al., 2011*). Other studies have documented lower levels of certain proinflammatory cytokines (e.g. IL-6, TNF-alpha) in *APOE4+* carriers (*Olgiati et al., 2010*), and downregulated expression of biomarkers mediating innate immune sensing (TLR-signaling molecules) (*Dose et al., 2018*).

Experimental studies also showed the associations of *APOE4* with heightened innate and complement inflammatory responses to lipopolysaccharide stimulation in *Gale et al., 2014*; *Tzioras et al., 2019*. Maintaining lower levels of baseline innate immunity may minimize the accumulative damage caused by low-grade innate inflammation over the long term, while still enabling strong targeted immune responses to pathogens following exposure (*Franceschi et al., 2000*; *Trumble and Finch, 2019*), particularly in environments with a diversity of species of pathogens. Further, pathogens like helminthic parasites, may moderate inflammation by triggering Th2-mediated (anti-inflammatory) immunological pathways (*Maizels and McSorley, 2016*; *Motran et al., 2018*), which may be important for counterbalancing a strong proinflammatory response.

In energy-limited, pathogenically diverse environments, *APOE4+* carriers may thus be better able to tolerate energetic costs imposed by infection by having higher concentrations of circulating lipids to fuel immune responses, while also minimizing damage from exposure to generalized systemic inflammation through downregulation of innate immune function. By contrast, in post-industrialized contexts, without the moderating influences of parasites on both cholesterol and immune functions, non-pathogenic stimuli (e.g. obesity) may be more likely to trigger systemic low-grade inflammatory pathways and, in the absence of a brake, lead to arterial and vascular damage and disease. Thus, the

link between *APOE4* and inflammatory diseases in post-industrialized contexts may in part be due to an environmental mismatch.

## Hypothesis and aims

In pathogenically diverse environments, innate immune defenses are likely to be activated owing to more frequent encounters with novel pathogens. This more diverse pathogenic setting may increase selective pressure to favor stronger immune regulation. We hypothesize that in such a context, an *APOE4* variant is less harmful because it (a) minimizes damage caused by chronic innate inflammation and (b) maintains higher circulating cholesterol levels, which buffer energetic costs of pathogen-driven innate immune activation. In post-industrial contexts, where there is a relative absence of diverse pathogens and thus reduced pathogen-mediated lipid regulation, coupled with an overabundance of calories, the effect of *APOE4* on circulating lipids may instead incur a cost. Lifestyle factors that promote obesity and excessive circulating lipids may lead to sterile endogenous inflammation (*Trumble and Finch, 2019*) that overshadows any potentially positive effects of *APOE4* on immune function. Thus, the *APOE4* variant has greater potential to lead to hyperlipidemia and coincide with related inflammatory diseases in high-calorie, low-pathogen, environments (*Figure 1*).

This study describes the immunophenotypes and lipid profiles of individuals with *APOE3/3* and *APOE4+* genotypes living in a pathogenically diverse, energy-limited environment. For the purpose of hypothesis testing, we focus on testing genotype-related differences in components of innate immunity (CRP, neutrophils, eosinophils, and erythrocyte sedimentation rate [ESR]) and blood lipids linked to inflammation (total cholesterol, LDL, and oxidized LDL [ox-LDL]). We evaluate the extent to which body mass index (BMI) moderates the association between lipids and innate inflammation (CRP, neutrophils, ESR). Finally, we test whether the *APOE4* allele has a moderating effect on the relationship between BMI and lipids to evaluate the role of *APOE4* in the maintenance of stable lipid levels under energetic restriction.

This research focused on the Tsimane, an Amerindian population in the Bolivian tropics that faces high exposure to a diverse suite of pathogens, and endemic helminthic infections. Tsimane have high rates of infection across all ages, with 70 % helminth prevalence and >50% of adults with co-infections from multiple species of parasite or protozoan (*Blackwell et al., 2015*; *Garcia et al., 2020*). Compared to U.S. and European reference populations, the Tsimane have also been found to have upregulated immune function across the life course (*Blackwell et al., 2016*). In most villages, there is little or no access to running water or infrastructure for sanitation (*Dinkel et al., 2020*). The Tsimane are primarily reliant on foods acquired through slash-and-burn horticulture, fishing, hunting, gathering, and small animal domestication, supplemented with market goods (e.g. salt, sugar, cooking oil) (*Kraft et al., 2018*). Tsimane are rarely sedentary, instead engaging in sustained low and moderate physical activity over much of their life course (*Gurven et al., 2013*), and have minimal atherosclerosis (*Kaplan et al., 2017*). However, with greater globalization and improvements in technology, the Tsimane are experiencing ongoing lifestyle changes; there is variation in participation in the market economy, and related variation in diet (e.g. access and uptake of processed foods) and activity level. Despite increasing access to markets and towns, infections remain the largest source of morbidity (*Gurven et al., 2020*).

## Results

Data include *APOE* genotype and multiple measurements of BMI, lipids, and immune markers in 1266 Tsimane adults. Mixed effects multiple regression models were used to accommodate for multiple measurements per individual for some biomarkers, as well as community-level differences in pathogen exposures. Leveraging multiple observations per individual over time should better capture individuals' average levels, and minimize the potential of single or few outliers driving results. Models also include covariates which adjust for age, sex, seasonality, and current infection (WBC >12 × 10⁹/L) (see Materials and methods for details). The sample is 50 % female and includes individuals from 80 villages. The mean ± SD age of the sample is 54 ± 11 years (range: 21–93). Mean ± SD BMI is 24 ± 3 for both sexes; Tsimane adults are relatively short (women: 150.5 ± 4.8 cm; men: 161.7 ± 5.3 cm) with low body fat (median body fat percentage for men and women is 18% and 26%, respectively). Prevalence of obesity is relatively low (10%). In general, blood immune biomarkers vary significantly across

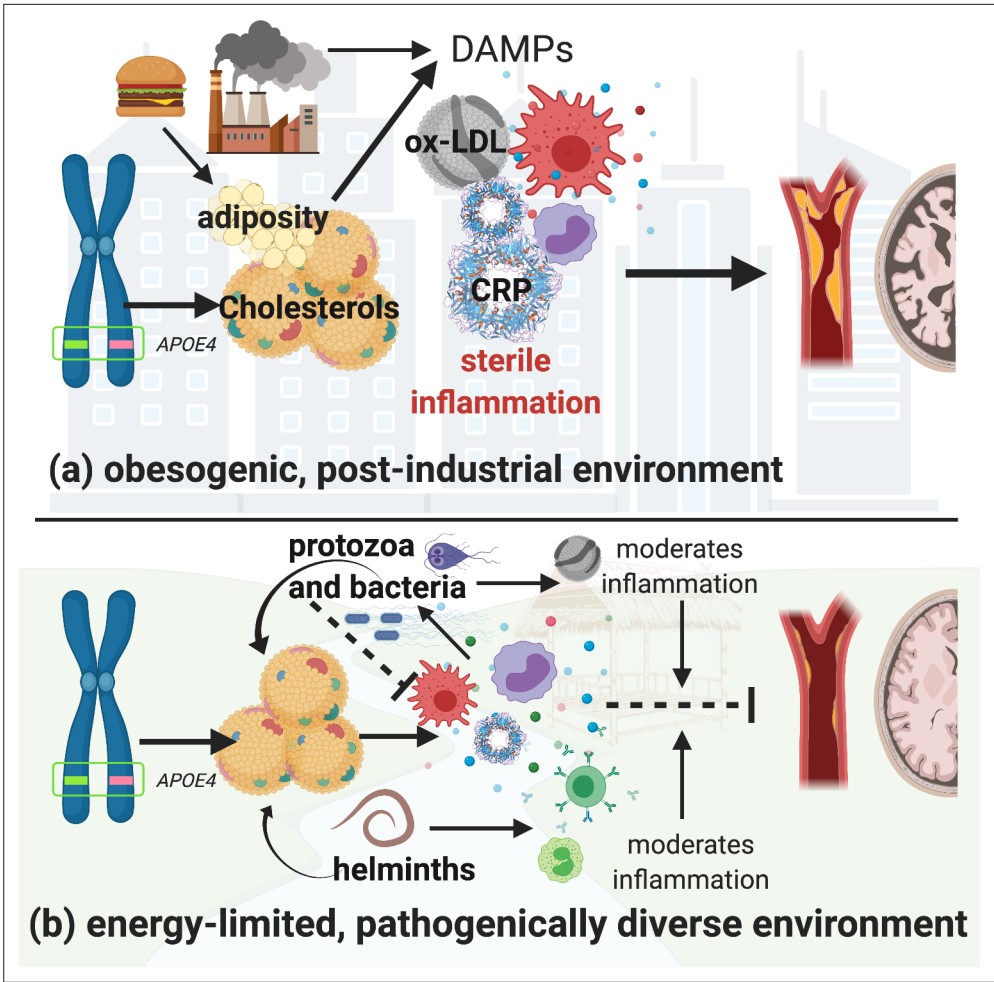

**Figure 1.** Hypothetical pathways through which the apolipoprotein *E4* (*APOE4*) allele influences lipid processing, immune regulation and disease risk in post-industrialized and non-industrialized contexts. In both contexts, the *APOE4* allele leads to increased levels of circulating lipids; however, in post-industrialized contexts (**a**), lipid levels can reach dangerously high levels due to obesogenic diets, and an absence of moderation by parasites and pathogen-driven immune activation. Immune activation by non-pathogenic elements triggers damage-associated molecular pattern pathways, which generates a proinflammatory 'sterile' immune response. Obesity and hyperlipidemia can simultaneously fuel sterile inflammation and promote oxidization of cholesterols, which, due to their lack of function, cause further tissue damage associated with cardiovascular and neurodegenerative disease risk. In energetically limited, pathogenically diverse contexts (**b**), the pathway between *APOE4* and disease risk is considerably more complex. Briefly, immune responses to parasites and microbes require cholesterol, and there are both direct and indirect effects of different species of parasites which further regulate cholesterol production and utilization. In addition, anti-inflammatory immune responses are generated by ox-LDL (e.g. in response to bacteria and protozoal infections), and helminthic parasites, which balance the immune system's overall response. It is possible that in contexts where there is higher pathogen diversity, an *APOE4* phenotype may be less harmful because it minimizes the damage caused by upregulated innate immune function, while also maintaining higher cholesterol levels which would buffer the cost of innate immune activation due to infection. Whereas in high-calorie, low-pathogen environments, the utility of having an *APOE4* allele may be muted, and the costs more severe. Image created with BioRender.com.

adult ages (*Supplementary file 1*; *Figure 2*). *Table 1* provides a full description of lipid and immune levels of individuals by *APOE* genotype.

In our sample, 21.2 % have at least one *APOE4* allele; 245 individuals are heterozygous *3/4*, and 23 are homozygous *4/4*. The remaining 78.8 % (*n* = 998) are homozygous for *APOE3/3*. Overall frequency of the *APOE4* allele is 11.5 %. The *APOE2* allele was absent. Throughout the rest of the paper, *APOE* genotype is defined categorically, binning individuals as homozygous *APOE3* (*E3/3*) or as *APOE4*+

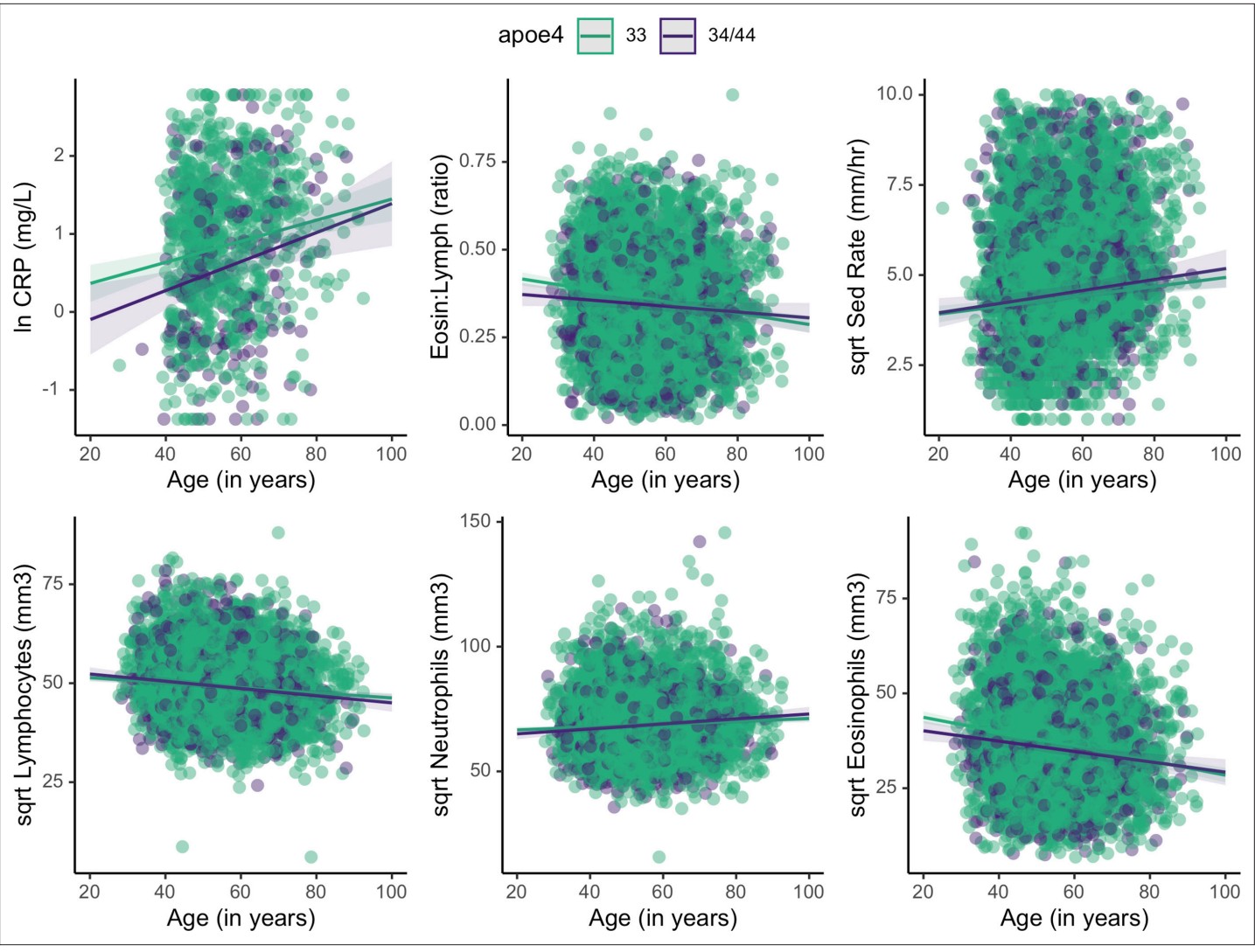

**Figure 2.** Plots showing estimated change in immune markers across age, split by apolipoprotein *E4* (*APOE*) genotype. Slopes were taken from mixed effects linear regression models and represent estimates of the interactive effects of *APOE* genotype and age on each immune marker. Models adjust for sex, season, and current illness (***Supplementary file 1***). Erythrocyte sedimentation rate is abbreviated as 'Sed Rate'.

carriers (if they had at least one copy of the *APOE4* allele). Heterozygous and *APOE4* homozygotes were binned together due to the small number of homozygotes (1.8 % of total sample, and thus too small of a group for adequately powered statistical tests).

## Characterization of lipid and immune profiles by APOE genotype

Relative to *APOE3/3* homozygotes, *APOE4* carriers have higher BMI ($\beta$ = 0.15 [CI: 0.02–0.28], p = 0.02), total cholesterol ($\beta$ = 0.15 [CI: 0.04, 0.27], p = 0.009), and oxidized LDL ($\beta$ = 0.16 [CI: –0.00, 0.32], p = 0.05), yet lower levels of innate immune blood biomarkers: CRP (β = –0.29 [CI: –0.44,–0.14], p < 0.001), eosinophils (β = –0.16 [CI: –0.24,–0.08], p < 0.001) (***Table 1***, ***Figure 3***). Similarly, in constrained models that excluded all individuals with CRP >10 mg/L and >5 mg/L, *APOE4* carriers maintained lower levels of CRP relative to *APOE3/3* homozygotes ($\beta$ = –0.23 [CI: –0.38,–0.09], p < 0.001; $\beta$ = –0.22 [CI: –0.36,–0.08], p = 0.002, respectively). Adjusting for multiple testing, CRP (FDR adj. p < 0.001) and eosinophils (FDR adj. p < 0.001) remain significantly different between *APOE* genotypes. *APOE4* is also associated with a lower eosinophil to lymphocyte ratio ($\beta$ = –0.14 [CI: –0.23,–0.05], p = 0.001) and lower total leukocytes ($\beta$ = –0.08 [CI: –0.15,–0.01], p = 0.02), but not with LDL, high-density lipoprotein (HDL), or triglycerides (***Figure 3—figure supplement 1***). Full models shown in ***Supplementary file 1b-e***.

**Table 1.** Description of immune and lipid measures for homozygous APOE3/3 and APOE4+ carriers for whom age and sex are available.

Values are reported as mean (standard deviation). Linear mixed effects models fit by REML were used to test for differences between groups, controlling for age, sex, current immune activation, and seasonality, with random effects for community and individual ID. Model to test age differences includes only a random effect for individual ID. Multiple test correction was conducted for models used for hypothesis-testing; False Discovery Rate (FDR) adjusted p-values are reported for these models. No. of observations is reported as: total number of observations for the main biomarker (number of unique individuals) in each model. T-statistics use Satterthwaite's method. Due to skewness of biomarkers, statistical models use transformed and scaled data for normalization, as such, estimates are reported as standardized betas (b) with standard errors (SE). Full models with covariates and 95% confidence intervals can be found in Tables 2-3 in the Supplement. See methods for transformations for each marker.

| Variables | APOE 3/3 | Apoe4+ | No. observations per model | β | SE | t-value | p-value | FDR adj p-value |
|---|---|---|---|---|---|---|---|---|
| N | 998 | 268 | – | – | – | – | – | – |
| % female | 51% | 48% | – | – | – | – | – | – |
| Age (in years) | 54 (11.5) | 54 (11.2) | 6615 (1268) | 0.007 | 0.07 | 0.106 | 0.915 | – |
| BMI (kg/m2) | 23.9 (3.4) | 24.6 (3.7) | 6224 (1263) | 0.15 | 0.06 | 2.421 | 0.016* | – |
| C-reactive protein (mg/L) | 3.7 (3.3) | 2.9 (2.6) | 1032 (907) | -0.29 | 0.08 | -3.731 | <0.001*** | <0.001*** |
| Eosinophil (mm3) | 1617 (1060) | 1375 (910) | 6129 (1259) | -0.16 | 0.04 | -3.871 | <0.001*** | <0.001*** |
| Neutrophil (mm3) | 5044 (1803) | 4816 (1697) | 6144 (1259) | -0.03 | 0.03 | -0.816 | 0.415 | 0.435 |
| Sed Rate (mm/hr) | 29.4 (19.5) | 29.7 (19.2) | 5987 (1253) | 0.06 | 0.04 | 1.334 | 0.182 | 0.217 |
| Eosin : Lymph ratio | 0.36 (0.14) | 0.33 (0.14) | 6121 (1260) | -0.14 | 0.04 | -3.197 | 0.001** | –--- |
| Leukocytes (1000/mm3) | 9.3 (2.5) | 8.8 (2.4) | 6229 (1266) | -0.08 | 0.03 | -2.281 | 0.023* | – |
| Lymphocytes (mm3) | 2590 (832) | 2565 (852) | 6136 (1260) | 0 | 0.04 | 0.104 | 0.917 | – |
| Hemoglobin (g/dL) | 13.2 (1.5) | 13.2 (1.4) | 6195 (1263) | -0.04 | 0.05 | -0.805 | 0.421 | – |
| Triglycerides (mg/dL) | 107 (51) | 114 (61) | 2633 (1174) | 0.07 | 0.06 | 1.157 | 0.248 | – |
| Total cholesterol (mg/dL) | 144 (32) | 148 (33) | 2581 (1168) | 0.15 | 0.06 | 2.594 | 0.010** | 0.021 |
| HDL cholesterol (mg/dL) | 38 (9) | 38 (9) | 2477 (1167) | 0.05 | 0.06 | 0.83 | 0.407 | – |
| LDL cholesterol (mg/dL) | 90 (32) | 93 (32) | 2390 (1154) | 0.08 | 0.06 | 1.307 | 0.192 | 0.217 |
| Oxidized LDL (U/L) | 76 (24) | 79 (23) | 1033 (907) | 0.16 | 0.08 | 1.947 | 0.052* | 0.090t |

Statistical significance is denoted as: t p<0.10; * p<0.05; ** p<0.01; *** p<0.001.

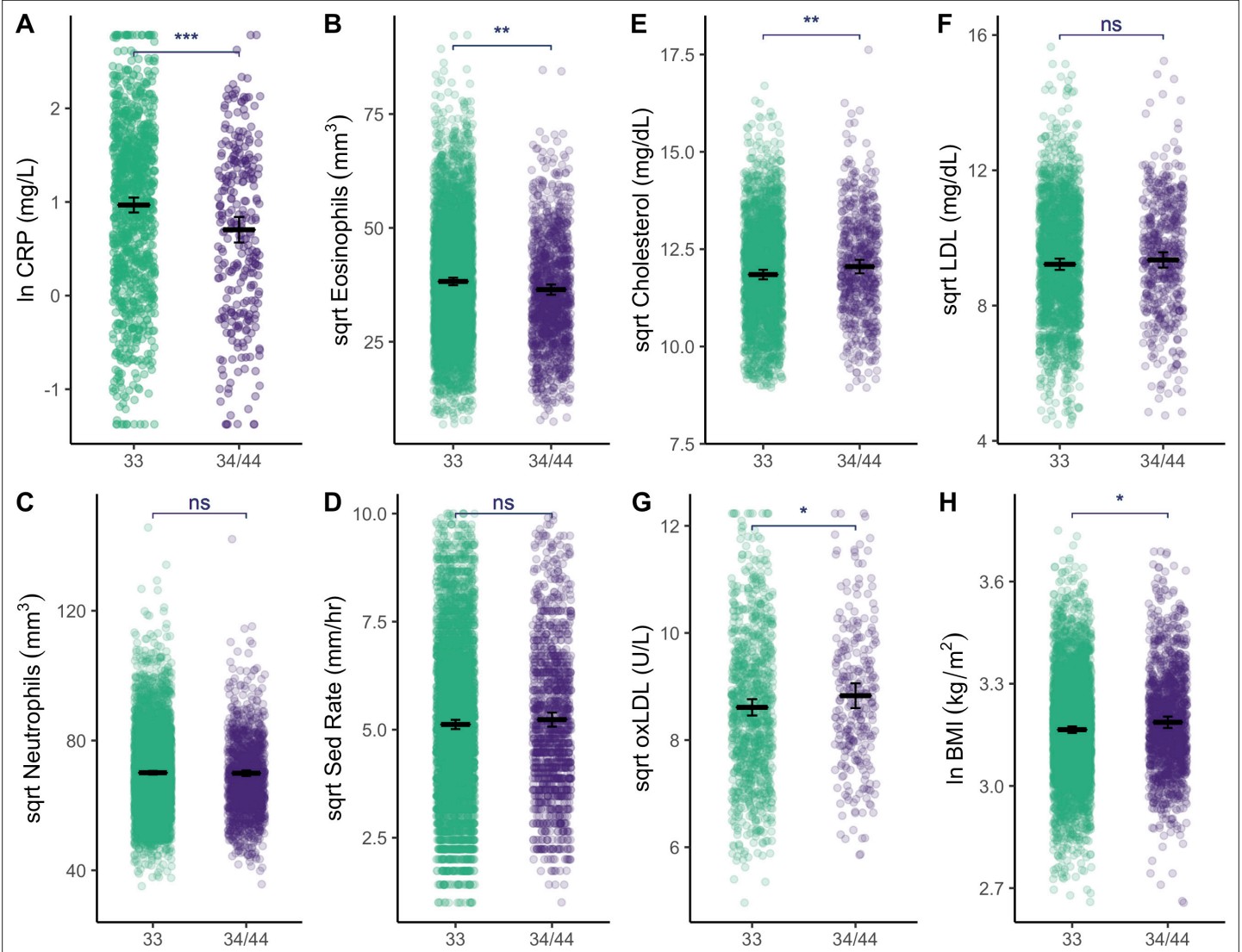

**Figure 3.** Plots show distributions of biomarker data (raw) grouped by APOE genotype (homozygous *APOE 33* versus those that have at least one copy of the *E4* allele). All plots include estimated means (horizontal lines) and standard errors (crossbars) per genotype, derived from mixed effects linear regression models that adjust for age, sex, season, and current illness. For full models with covariates, see ***Supplementary file 1b, c and e***. Erythrocyte sedimentation rate is abbreviated as 'Sed Rate'. See ***Figure 3—figure supplement 1*** for additional plots comparing of biomarkers by APOE genotype. Statistical significance is denoted as: *p ≤ 0.05; **p ≤ 0.01; ***p ≤ 0.001.

The online version of this article includes the following figure supplement(s) for figure 3:

**Figure supplement 1.** Plots show distributions of biomarker data (raw) grouped by APOE genotype (homozygous *APOE 33* versus those that have at least one copy of the *E4* allele).

### Does BMI moderate the association between lipids and inflammation?

For Tsimane with higher (>28) BMI, total cholesterol and LDL did not associate with CRP; however, for individuals with median (21 ≤ BMI ≤ 28) or low (<21) BMI, higher total cholesterol and LDL associate with lower CRP (***Figure 4***). When considered as a continuous variable, BMI significantly moderates these associations (total cholesterol: $\beta = 0.15$ [CI: 0.08, 0.21], $p < 0.001$; LDL: $\beta = 0.16$ [CI: 0.10, 0.22], $p < 0.001$). BMI also interacts with ox-LDL ($\beta = 0.14$ [CI: 0.08, 0.19], $p < 0.001$) in predicting CRP; however, this relationship is distinct from the other lipids tested. For Tsimane with high BMI, ox-LDL positively associates with CRP, while for those with low BMI the inverse is true (***Figure 3C***). For ESR, total cholesterol ($\beta = 0.05$ [CI: 0.01, 0.08], $p = 0.008$) and LDL ($\beta = 0.03$ [CI: –0.00, 0.07], $p = 0.07$) positively associate with ESR only among Tsimane with higher BMIs. After adjusting for multiple testing,

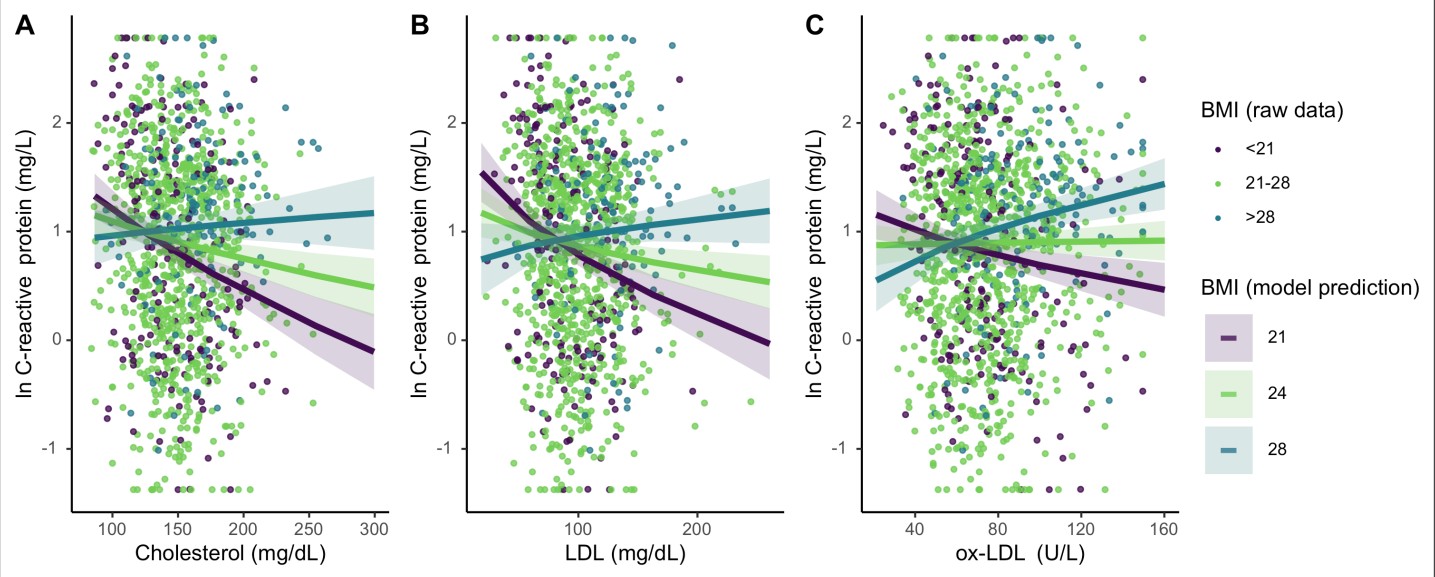

**Figure 4.** Differing influence of cholesterols on C-reactive protein (CRP), based on three levels of body mass index (BMI) (mean, ± 1 standard deviation). Panels A-C show interaction effects between BMI and (**A**) total cholesterol, (**B**) LDL, and (**C**) oxidized LDL. For total and LDL cholesterol, among those with low (purple line) and mean (lime green line) BMI, cholesterol is negatively associated with CRP. Oxidized LDL is only associated with higher CRP among individuals with high BMI (turquoise line). Lines are predicted from mixed effects linear regressions that adjust for age, sex, seasonality, with random effects for individual and community residence (***Supplementary file 1f-h***). Data points are raw values, color-coded based on BMI level. Variables are transformed and centered. See ***Figure 4—figure supplement 1*** for independent associations between BMI and cholesterols with markers of inflammation.

The online version of this article includes the following figure supplement(s) for figure 4:

**Figure supplement 1.** This figure shows independent associations between body mass index (BMI) and cholesterols with markers of inflammation from mixed effects models, adjusting for age, sex, seasonality, and current illness with random effects for community and individual ID.

relationships between cholesterols and CRP all remain significant (all FDR adj. p < 0.001), as does the relationship between total cholesterol and ESR (FDR adj. p = 0.02).

Concerning independent relationships, there are no direct relationship between ox-LDL and CRP, whereas higher total cholesterol and LDL are associated with lower CRP (total cholesterol: $\beta$ = –0.13 [CI: −0.20,–0.05], p < 0.001; LDL: $\beta$ = –0.11 [CI: −0.18,–0.03], p = 0.006) (***Figure 4—figure supplement 1***). Total cholesterol is also negatively associated with ESR ($\beta$ = –0.06 [CI: −0.10,–0.02], p = 0.002). There are no independent or interactive relationships between BMI, cholesterols, and neutrophils. For full models, see ***Supplementary file 1f-h***.

## Does *APOE* genotype moderate the association between BMI and lipids?

Finally, we assess whether *APOE* genotypes differentially moderate associations between cholesterol and BMI for lean and high BMI individuals. To evaluate the effects of the *APOE4* allele on lipid levels across the range of BMI, we added an interaction term between BMI and *APOE* genotype to the mixed effects linear regression models assessing relationships between BMI and lipids (***Table 2***). These analyses show that *APOE4* carriers maintain similar levels of total cholesterol and LDL across BMIs (cholesterol: $\beta$ = –0.08 [CI: −0.18,–0.02], p = 0.11; LDL: $\beta$ = –0.09 [CI: −0.19, 0.00], p = 0.06), whereas *APOE3/3* homozygotes show higher cholesterol with BMI. Specifically, *APOE4* carriers maintain higher levels of total and LDL cholesterol at lower BMIs, but have lower levels of both at higher BMIs, relative to individuals that are homozygous *APOE3/3* (***Figure 5***). However, neither of these associations are significant. Ox-LDL, HDL, and triglycerides do not vary by *APOE* alleles across BMIs. For full models, see ***Supplementary file 1***.

**Table 2.** Models evaluating the moderating effects of APOE genotype on associations between BMI and cholesterols.

Results are fixed effects estimates from mixed effects linear regressions, which include random effects for ID and community residence. In addition to age, sex, and season, a dummy variable was used as a proxy for current illness (leukocytes > 12 mm3). Results are reported as standardized betas; CI is the 95% confidence interval. All dependent variables were transformed and centered prior to analyses. APOE genotype is coded as a categorical variable, binned as individuals that are homozygous E3 (E3) versus those that have at least one copy of the E4 allele (E4).

| Predictors | Total cholesterol | | | LDL | | | Oxidized LDL | | |
|---|---|---|---|---|---|---|---|---|---|
| | β | CI | p | β | CI | p | β | CI | p |
| E4 * BMI | −0.08 | −0.18–0.02 | 0.111 | −0.09 | −0.19–0.00 | 0.061 | 0.02 | −0.13–0.17 | 0.786 |
| APOE [E4] | 0.14 | 0.03–0.26 | 0.013 | 0.07 | −0.04–0.18 | 0.221 | 0.13 | −0.03–0.29 | 0.112 |
| BMI | 0.18 | 0.14–0.23 | <0.001 | 0.2 | 0.15–0.25 | <0.001 | 0.21 | 0.14–0.28 | <0.001 |
| Sex [women] | 0.15 | 0.06–0.24 | 0.001 | 0.2 | 0.11–0.28 | <0.001 | 0.09 | −0.03–0.21 | 0.158 |
| Age [in years] | 0.01 | 0.01–0.01 | <0.001 | 0.01 | 0.01–0.02 | <0.001 | 0 | −0.01–0.01 | 0.869 |
| Currently ill | 0.02 | −0.13–0.17 | 0.773 | −0.03 | −0.18–0.13 | 0.744 | 0.15 | −0.09–0.39 | 0.229 |
| Season [wet] | −0.12 | −0.20–−0.04 | 0.004 | −0.29 | −0.38–−0.21 | <0.001 | -0.28 | −0.44–−0.13 | <0.001 |
| (Intercept) | −0.7 | −0.94–−0.46 | <0.001 | −0.71 | −0.95–−0.47 | <0.001 | 0.09 | −0.28–0.46 | 0.626 |
| Observations | 2581 | | | 2390 | | | 1032 | | |
| Marginal R2 | 0.049 | | | 0.074 | | | 0.07 | | |

## Discussion

We hypothesized that the effect of the *APOE4* polymorphism on disease risk may be environmentally moderated, and that in an energy-limited and pathogenically diverse context, carrying an *APOE4* allele may provide some benefit if it aligns with downregulated innate immune function and higher circulating lipids. In support of this hypothesis, we find that Tsimane with an *APOE4* allele have higher levels of lipids, yet lower levels of CRP and eosinophils, compared to individuals that have a homozygous *APOE3* genotype. Overall, *APOE4* carriers also have an adaptive-biased immune profile (as measured by a lower ratio of eosinophils to lymphocytes). These associations remain when controlling for factors known to contribute to differences in inflammation including age, sex, seasonality, and current infection. The most striking example of immunological differences is for CRP, an acute phase protein that functions as part of the complement system and is a marker of generalized inflammation. Though CRP is higher in the Tsimane relative to U.S. and European populations (*Blackwell et al., 2016*; *Gurven et al., 2008*), we find that CRP levels are 30 % lower among *APOE4* carriers than individuals that are homozygous *APOE3/3*. This replicates a similar finding of lower CRP among Tsimane *APOE4* carriers from samples collected over a decade prior to the current study (*Vasunilashorn et al., 2011*). A study of 'healthy' (nondiabetic) Finnish men also found that *APOE4* carriers had higher LDL cholesterol coupled with lower CRP (*Martiskainen et al., 2018*).

Further, in support of our hypothesis that lifestyle and ecological (e.g. pathogen exposure) factors affect how lipids influence inflammation, we find that BMI significantly moderates relationships between lipids and markers of innate inflammation. Our results are consistent with expectations for a high infection context: Tsimane with high BMI (~1 SD above the mean) show no relationship between total cholesterol or LDL with CRP and ESR, but those with mean or low BMI have *higher* lipid levels that are associated with *lower* CRP and ESR. In contemporary post-industrialized contexts (*Figure 1a*), these findings may seem counterintuitive. But in a high-pathogen context, higher concentrations of circulating lipids may allow individuals, especially those that are energy-compromised, to better tolerate infection (*Gurven et al., 2016*). Thus, the negative association between cholesterol and inflammatory markers could indicate lower infectious burden. For the Tsimane, higher cholesterol levels are likely an indicator of robust health (e.g. not fighting an infection) rather than chronic disease. Independently, lipids are either neutral or negatively associated with immune markers in this sample. This is consistent

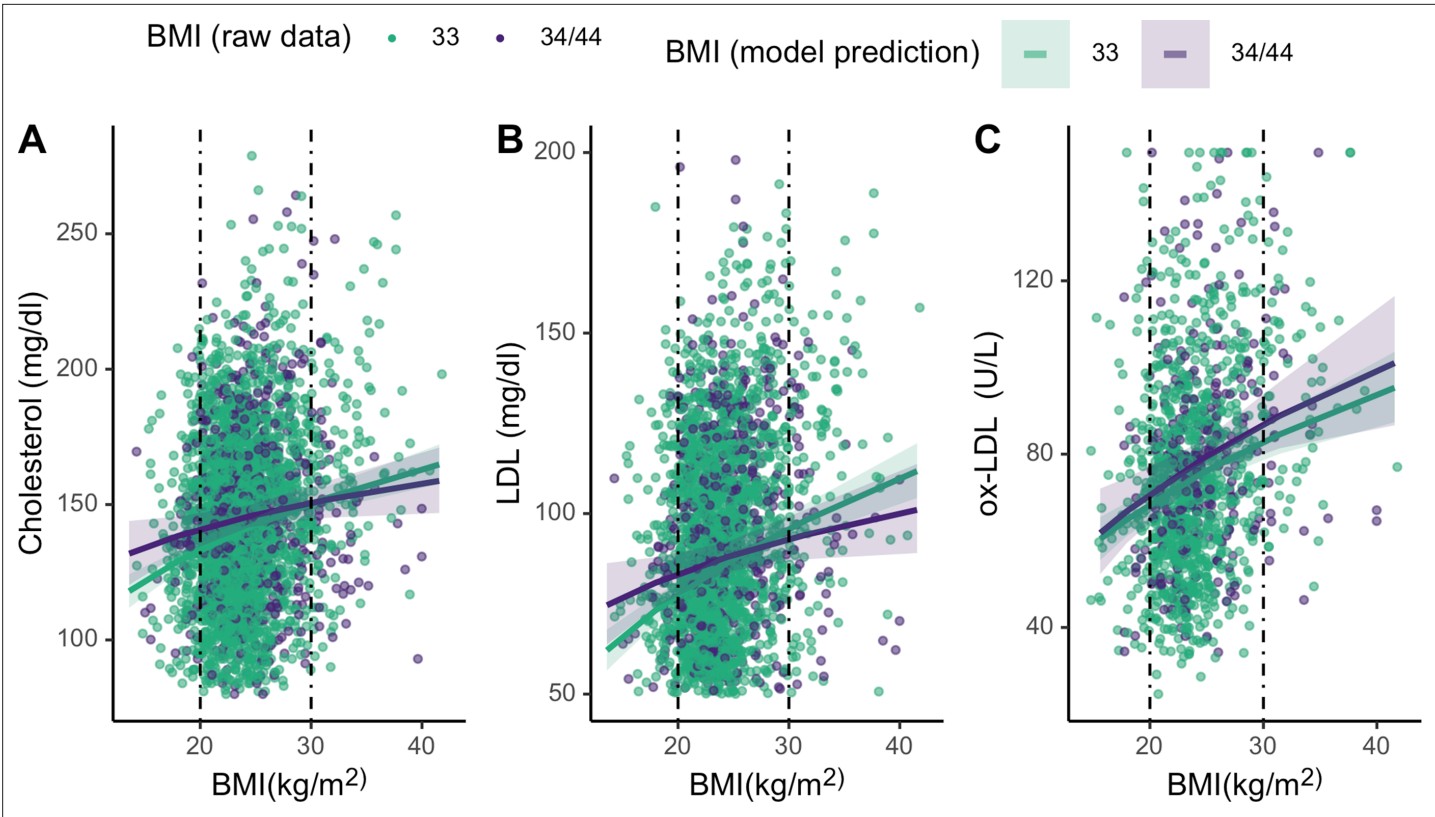

**Figure 5.** Plots showing moderating effects of *APOE* genotype on associations between body mass index (BMI) and cholesterols (see Table 2 in the manuscript for models). Dotted vertical lines represent cutoffs for low (<21) and high (>28) BMI. For the primary cholesterols utilized during an immune response to pathogens – total cholesterol (**A**) and LDL (**B**) – individuals with an *E4* allele maintain slightly higher levels of those cholesterols at low BMIs, compared to homozygous *E3*. Plotted lines are derived from mixed effects linear regressions that include age, sex, season, and current infection as covariates, as well as random effects for individual and community residence. Data points are raw values, color-coded by *APOE* genotype.

with previous research among the Tsimane that found blood lipids varied inversely with IgE, eosinophils, and other markers of infection (*Vasunilashorn et al., 2010*). Overall, the Tsimane maintain low levels of cholesterol relative to U.S. and European standards – less than 5 % of sample had a total cholesterol higher than 200 mg/dL. The lack of hyperlipidemic cholesterol levels likely also explains in part why cholesterol does not associate with higher inflammation in this population.

Notably, the moderation effect of BMI indicates that the relationship between ox-LDL and CRP are inverted at the low and high ends of the BMI range. For individuals with low BMI, higher ox-LDL levels are associated with lower CRP, while for those with high BMI, higher ox-LDL is associated with higher CRP (*Figure 3c*). One possible explanation for the opposing relationships we find between ox-LDL and CRP is that there may be differences in the underlying causes of oxidization in these groups. Though ox-LDL is often considered a consequence of obesogenic or hyperlipidemic oxidization of LDL (*Neuparth et al., 2013*), ox-LDL is also produced in response to pathogen-mediated immune activation (*Han, 2009*), and its relationship with downstream immune functions largely depends on the cause of oxidation. For example, while nonpathogenic oxidization of LDL promotes inflammation (*Neuparth et al., 2013*; *Tall and Yvan-Charvet, 2014*), during protozoal or bacterial infections, ox-LDL contributes to lower inflammatory responses (*Han, 2009*). Thus, those with high BMI may represent the transition to sterile inflammation as lifestyles change with greater participation in a market economy (*Trumble and Finch, 2019*). Indeed, other studies have documented what may be the start of a nutritional transition among the Tsimane (*Kraft et al., 2018*; *Masterson et al., 2017*; *Rosinger et al., 2013*).

Our finding that innate immune biomarkers are lower among *APOE4* carriers is in line with prior reports (*Lumsden et al., 2020*; *Martiskainen et al., 2018*; *Trumble et al., 2017*; *Vasunilashorn et al., 2011*), however the causes are uncertain. One proximate explanation involves the mevalonate

pathway, which plays a key role in multiple cellular processes, including modulating sterol and cholesterol biosynthesis and innate immune function (*Buhaescu and Izzedine, 2007*). One ultimate explanation that we consider relates to the utility of downregulated baseline inflammation in contexts where there is likely chronic exposure to a wide diversity of novel pathogens. In such a context, the costs of inflammation would be amplified, and elements that are involved in regulating innate inflammation may be under stronger selective pressure. This would be plausible so long as downregulating immune function does not compromise one's 'ability to respond' to immunological threats, as has previously been argued (*Man et al., 2016*). Regarding the main finding for CRP, it is also possible that this is because *APOE4* carriers experience a lower innate immune sensing (*Dose et al., 2018*) or have faster clearance following the resolution of an acute spike. While there is currently no direct evidence for the latter, some studies have found that higher circulating lipids were associated with more rapid clearance of active infections (*Andersen, 2018*; *Pérez-Guzmán et al., 2005*). The current study design did not allow analysis of these potential pathways.

Though numerous other studies have established links between the *APOE4* polymorphism and neighboring genes (the *APOE* gene cluster) (*Kulminski et al., 2019*), and increased circulating lipids (*Yassine and Finch, 2020*; *de Chaves and Narayanaswami, 2008*; *Safieh et al., 2019*; *Saito et al., 2004*), to our knowledge, this study is the first to assess whether *APOE* moderates lipid levels under energetic constraints. Specifically, our results show that the *APOE4+* genotype is marginally associated with higher relative lipids in a low-energy, pathogenically diverse system. In one study conducted in an obesogenic environment (mean BMI 28 kg/m$^2$), BMI was independently associated with higher total cholesterol, but this relationship did not differ by *APOE* genotype (*Petkeviciene et al., 2012*). While it is difficult to pinpoint the cause for discrepant findings across studies, one possibility is that the Lithuanian study did not capture moderation at low BMI (the mean BMI was 28 for both *E4 APOE4+* and *APOE3/3* genotypes). Another possibility is that the low-pathogen context of urban Lithuania led to less moderation of cholesterols at high BMIs. As mentioned in Introduction, one proposal for the persistence of the ancestral *E4* allele despite its deleterious health effects at later ages relies on the fitness benefits of lipid buffering in early life relative to the more recent *E3* mutation (*van Exel et al., 2017*; *Yassine and Finch, 2020*). The ability to maintain adequate lipid reserves under energetic and pathogenic pressures would also provide additional benefits over the life course (*Finch and Kulminski, 2021*).

Finally, nearly one-fourth of individuals in the sample had at least one copy of *APOE4*. The allelic frequency of *APOE4* reported here is similar to what has been previously documented among other South American and Amerindian populations, which ranges from 5% to 30% and varies widely by region and level of genetic admixture (*Corbo and Scacchi, 1999*; *Gayà-Vidal et al., 2012*). The distribution of *APOE* allelic variants in populations around the world is likely due to a mosaic of factors, from genetic drift, to antagonistic pleiotropy, to potential differences related to environmentally dependent costs and benefits of functionally distinct variants. Certainly some degree of genetic drift or founder effects (*Gayà-Vidal et al., 2012*; *Singh et al., 2006*), antagonistic pleiotropy (*Smith et al., 2019*; *van Exel et al., 2017*) and other forces (e.g. genetic relatedness) play an important role in determining population and global frequencies and should be considered jointly to make inferences about the frequency of *E4* in the Tsimane population. While pathogen risk and energetic limitations is one context where the typical costs of *E4* may not be expressed, the high frequency of *E4* allele in populations that are appreciably different in terms of latitude, ecology, and population history (e.g. northern European and central African populations) confirms that more is involved in understanding population differences in *E4* frequency (*Abondio et al., 2019*; *Eisenberg et al., 2010*).

The commonly reported associations between *APOE4* and disease risk in post-industrialized societies may differ from subsistence populations due to environmental mismatch (*Trotter et al., 2010*). In an environment where calories are limited, and one's innate immune system is already primed by multiple pathogens, the benefit of having an *APOE4* allele – which facilitates upregulated lipids (sufficient for mounting immune responses) and downregulates innate immune function – may be amplified. Such a phenotype could be beneficial particularly if downregulated immune function did not compromise responses to immunological threats. Further, in such an environment, this phenotype may be energetically conservative, maintaining resources to fuel other systems, such as cognitive functioning (*Trumble and Finch, 2019*). Indeed, there is some evidence that *APOE4* carriers may be better able to buffer the negative effects of infection. An *APOE4* polymorphism in Brazilian children with a

history of infection (measured by bouts of diarrhea in infancy) predicted better scores on cognitive exams (*Oria et al., 2011*). Among Tsimane infected with parasites, only *APOE4* carriers performed better on cognitive tests (*Trumble et al., 2017*). However, in hygienic post-industrialized contexts, the risk of helminthic and protozoal infection is low, and most pathogens can be quickly treated with medication. Thus, the potential benefits of the *APOE4* allele are muted, and the costs of higher lipids are much more severe in these obesogenic environments.

## Limitations

Though our findings draw from a large sample size and are robust to various controls and model specifications, there are several limitations. First, our findings are correlative and limit causal inference. Because these findings may be important for furthering evolutionary (i.e. why the *APOE4* allele may be maintained) and clinical (i.e. the role of *APOE* in disease pathogenesis) understanding, they require replication and warrant experimental testing. The central thesis presented here – that persistent exposure to pathogens and obesogenic diets moderate the relationship between blood lipids and inflammation – is amenable to experimental manipulation under lab conditions. Specifically, a mammalian model system could be split into two treatments: those raised under sterile conditions versus regimented exposure to non-lethal pathogens. These treatments may then be crossed with dietary or physical activity conditions that produce differential levels of adiposity. Our hypothesis predicts that both decreased adiposity and increased life course pathogen exposure will reduce or even eliminate positive associations between blood lipids and chronic inflammation. Importantly, inflammatory biomarkers can be measured at more frequent intervals in lab conditions to assess long-term differences in the function of both pro- and anti-inflammatory pathways between experimental treatments.

Second, there is some evidence that the *APOE4* allele is positively associated with HDL cholesterol levels (*Hopkins et al., 2002*), and that higher HDL levels reduce risk of severe infection (measured by infectious hospitalizations) (*Trinder et al., 2020*). Inversely, acute infections and systemic inflammation (e.g. acute phase reaction) are associated with a decrease in HDL and HDL remodeling that results in lower cholesterol efflux capacity and higher peripheral levels of LDL cholesterol (*Ronsein and Vaisar, 2017*; *Zimetti et al., 2017*). While we did not find differences in HDL by *APOE* status among the Tsimane, we cannot completely discount the possibility that HDL remodeling plays a role in the higher lipid levels, in addition to *APOE* allelic variation. Further, given the relative lack of, and difficulty in accessing, medical care facilities, it is difficult to assess degrees of infection severity, and thus it is also possible that *APOE4* may mitigate infectious disease burden. However, the current data cannot provide evidence for either of these potentials, and further research is needed to disentangle the roles of *APOE* and lipids in infection.

Third, proxies for energy availability and pathogen exposures are imperfect. BMI and fat free mass may have a variable relationship across the degree of market integration. However, given that our main goal in the paper – with regard to energy availability – is to investigate *APOE* and lipids at the extreme tails of BMI (lean versus obese), BMI should adequately capture broad differences in energetic availability between these two groups. Because patterns of immune response vary depending on pathogen type and species, the use of a high white blood cell count cutoff as a proxy for current infection overly simplifies immune variation due to different types of infection. To this end, we also adjust for seasonality and cluster by community residence in our models, which should capture additional variation in exposures.

Finally, we were not able to fully adjust for the time of day that samples were collected for the biomarkers used in this paper. Given that peripheral levels of most immune biomarkers vary diurnally to some extent, it is possible that not adjusting for exact time of day may have introduced some noise into analyses. However, CRP (which the main findings centered around) does not appear to follow a circadian rhythm in healthy individuals (*Meier-Ewert et al., 2001*). Further, the largest differences in levels (peak to trough amplitudes) tend to coincide with sleep and wake cycles (*Labrecque and Cermakian, 2015*). Because blood draws routinely occur in the morning, samples are constrained to a narrow window, and thus we are not comparing values across the full range of diurnal variation.

## Conclusion

In post-industrial settings, *APOE4* is generally considered a purely deleterious allele, increasing inflammation and lipids, and escalating CVD and neurological disease risk. Yet, in a high-pathogen

environment with minimal obesity, we find that *APOE4* is associated with *lower* levels of innate inflammation. While *APOE4* carriers do have higher lipid levels, these may be beneficial for immune response and child survival, and unlikely to increase CVD risk in a population without other cardiometabolic risk factors.

## Materials and methods

### Population and sampling design

The Tsimane are an Amerindian population that live in the tropical lowlands of Bolivia. As of 2015, the Tsimane Health and Life History Project (THLHP) census estimated a total population size of about 16,000 individuals living across 90+ villages (*Gurven et al., 2017*). Data come from the THLHP, a longitudinal study of health and behavior that has run continuously since 2002 (*Gurven et al., 2017*).

### Sampling design

Biomarker data used in this paper were collected by the THLHP between 2004 and 2015 (see *Gurven et al., 2017*; *Kraft et al., 2020* for details). A Bolivian and Tsimane mobile medical team travel annually or biannually among study communities conducting clinical health assessments and collecting biochemical and anthropometric information from community members that want to participate. This sample includes all data from individuals for whom we have *APOE* genotyping and measurements of age, sex, and BMI – which is the base criteria for this study. Sample size varies by biomarker and over time for several reasons: sampling strategy varies by data type, absent or sick team personnel needed to collect data, the number of study villages and thus enrolled participants has increased over time, and the data types collected have changed over time (see *Kraft et al., 2020*). There are also fewer repeat measurements for a subset of biomarkers (i.e. CRP and ox-LDL) that were assayed in the United States, due to them being analyzed as part of a prior project. Specific sample sizes are reported in *Table 1*, and full tables report sample size for each model.

### APOE genotyping

Whole blood samples were stored in cryovials (Nalgene, Rochester, NY) and frozen in liquid nitrogen before transfer on dry ice to the University of California-Santa Barbara, where they were stored at –80 °C until genotyping. SNP genotyping was used to identify *APOE* allelic variants in blood samples. Samples were shipped on dry ice to University of Southern California (2010 and 2013) and University of Texas-Houston (2016), where DNA was extracted, quantified, and haplotype-coded for *APO- E2*, *E3*, and *E4* alleles using the TaqMan Allelic Discrimination system (Thermo Fisher Scientific, Carlsbad, CA). Determination of the *APOE2/E3/E4* alleles in the Tsimane derived from two SNPs of 20–30 bp oligonucleotides surrounding the polymorphic site (Cys112Arg/rs429358 and Cys158Arg/rs7412) (*Trumble et al., 2017*; *Vasunilashorn et al., 2011*).

### Measurement of blood lipids and immune function

Biomarkers were assayed either in the field at the time of collection or in the Human Biodemography laboratory at UC Santa Barbara in 2016.

Blood was collected by venipuncture in a heparin-coated vacutainer. Immediately following the blood draw, total leukocyte counts and hemoglobin were determined with a QBC Autoread Plus dry hematology system (QBC Diagnostics), with a QBC calibration check performed daily to verify QBC performance. Relative fractions of neutrophils, eosinophils, and lymphocytes were determined manually by microscopy with a hemocytometer by a certified Bolivian biochemist. ESR was calculated following the *Westergren, 1957*, method.

Serum was separated and frozen in liquid nitrogen before transfer to the University of California-Santa Barbara where a commercial immunoassay was used to measure ox-LDL (Mercodia Cat# 10-1143-01, Winston Salem, NC). Serum high sensitivity C-reactive protein (hs-CRP) was assessed via immunoassay (*Brindle et al., 2010*) and was cross-validated by the University of Washington laboratory, using the protocols utilized for the National Health and Nutrition Evaluation Survey (NHANES) (Meridian Life Sciences Cat# M86005M, RRID:AB_150654). Ox-LDL and hs-CRP assays use materials from the same lot across all measures. Total and LDL cholesterol levels from serum samples were

measured (Stat Fax 1908, Awareness Technology, Palm City, FL) in the THLHP laboratory in San Borja, Beni, Bolivia.

## Age estimation and anthropometrics

Birth years were assigned based on a combination of methods including using known ages from written records, relative age lists, dated events, photo comparisons of people with known ages, and cross-validation of information from independent interviews of kin (*Gurven et al., 2007*). Each method provides an independent estimate of age, and when estimates yielded a date of birth within a 3-year range, the average was used. Individuals for whom reliable ages could not be ascertained are not included in analyses.

Weight and height were measured in the field by a member of the THLHP medical team, using a basic digital scale (Tanita, Arlington Heights, IL) and stadiometer to the nearest 0.1 cm. BMI was calculated as weight (kg)/height$^2$ (m$^2$).

## Statistical analysis

Mixed effects linear regressions with restricted maximum likelihood estimation are used for all analyses. Models adjust for age, sex, seasonality, and current infection (leukocyte count >12,000 cells per microliter of blood) (*McKenzie and Williams, 2010*), with random intercept effects for individual ID and community. Because Tsimane villages vary in sanitation infrastructure, including access to soap and other hygienic products, and potentially prevalence by pathogen type (e.g. some living very close to the river versus farther out in the forest), individuals were clustered by community to account for variation in such community-level factors. Household-level variation (e.g. cohabiting individuals may have similar exposures) was also tested by including a random household ID as a random variable. However, because inclusion of household ID as a random effect did not alter parameter estimates or p-values, and fit (measured by AIC) was not substantially improved across any model, it was omitted from final analyses. To model moderation effects (sections Does BMI moderate the association between lipids and inflammation? and Does *APOE* genotype moderate the association between BMI and lipids?) interaction terms are included between the main predictor and moderator.

Immune and lipid measures required transformation to normalize their skewed distributions. Variables were transformed as follows: CRP, BMI, and triglycerides were natural log-transformed; total leukocytes and subsets (lymphocytes neutrophils, eosinophils), ESR, and remaining cholesterols (total cholesterol, LDL, HDL, ox-LDL) were square root-transformed. To compare across models, all dependent variables were then z-scored for analyses, and thus all betas are standardized estimates.

## Acknowledgements

We thank Tsimane participants and their communities, and the THLHP field team (including administrators, logistical support, physicians, biochemists, and anthropologists), whose support, expertise, and commitment made this work possible.

## Additional information

### Funding

| Funder | Grant reference number | Author |
|---|---|---|
| National Institute on Aging | R01AG024119 | Michael D Gurven Hillard Kaplan |
| National Institute on Aging | R56AG024119 | Michael D Gurven |
| Agence Nationale de la Recherche | | Jonathan Stieglitz |

The funders had no role in study design, data collection and interpretation, or the decision to submit the work for publication.

## Author contributions
Angela R Garcia, Conceptualization, Data curation, Formal analysis, Investigation, Methodology, Software, Visualization, Writing - original draft, Writing – review and editing; Caleb Finch, Conceptualization, Supervision, Writing – review and editing; Margaret Gatz, Kenneth Buetow, Supervision, Writing – review and editing; Thomas Kraft, Data curation, Methodology, Software, Writing – review and editing; Daniel Eid Rodriguez, Investigation, Writing – review and editing; Daniel Cummings, Data curation; Mia Charifson, Bret A Beheim, Data curation, Writing – review and editing; Hooman Allayee, Formal analysis, Resources, Writing – review and editing; Gregory S Thomas, Funding acquisition, Writing – review and editing; Jonathan Stieglitz, Funding acquisition, Project administration, Writing – review and editing; Michael D Gurven, Funding acquisition, Resources, Supervision, Writing – review and editing; Hillard Kaplan, Funding acquisition, Project administration, Resources, Supervision, Writing – review and editing; Benjamin C Trumble, Conceptualization, Funding acquisition, Investigation, Project administration, Resources, Supervision, Writing – review and editing

## Author ORCIDs
Angela R Garcia http://orcid.org/0000-0002-6685-5533
Caleb Finch http://orcid.org/0000-0002-7617-3958
Margaret Gatz http://orcid.org/0000-0002-1071-9970
Hooman Allayee http://orcid.org/0000-0002-2384-5239
Jonathan Stieglitz http://orcid.org/0000-0001-5985-9643
Michael D Gurven http://orcid.org/0000-0002-5661-527X

## Ethics
Human subjects: This research has been approved by institutional review boards at the University of New Mexico (#07-157) and University of California Santa Barbara (#3-20-0740), as well as the Tsimane government (Tsimane Gran Consejo) and village leaders. Study participants give consent for each part of the research and data collection prior to participating, during every visit by the THLHP.

## Decision letter and Author response
Decision letter https://doi.org/10.7554/eLife.68231.sa1
Author response https://doi.org/10.7554/eLife.68231.sa2

---

# Additional files

## Supplementary files
• Transparent reporting form

• Supplementary file 1. Full models with covariates described in analyses and figures presented in main article.

• Supplementary file 2. Spanish-language abstract. Link to the Spanish version of the article: https://tsimane.anth.ucsb.edu/results.html#pubs.

## Data availability
Working with indigenous populations requires special care and sensitivity, especially in regards to the public sharing of genetic data. A long history of abuse and mistrust between researchers and indigenous populations behooves us to prioritize the wishes of Tsimane. Given current ethical restrictions on sharing Tsimane genetic data, we cannot upload Tsimane genetic data to a public server at this time. Individuals interested in accessing the APOE genetic data should submit a request to the co-senior authors. Access will be granted for health focused research, subject to approval by THLHP co-Directors and Tsimane Gran Consejo (Tsimane governing body). All other de-identified data and code are openly accessible on Dryad through the following link: https://datadryad.org/stash/share/FqfEp93SdnqklFbfbjBbenHsZh_10gulo8cf_9LMR3o While we are strong advocates of open science and believe that inclusion of diverse populations in genomics is critical for advancing the goals of public health and health equality, we also need to ensure that all research participants are protected. Our research team is working toward a long-term solution for data-sharing and a streamlined process for requests. This includes the creation of a locally-based ethics and review board comprised of representatives from Tsimane communities and leadership, research collaborators at the Universidad de San Simon (Cochabamba), and THLHP co-directors. The goal of this board is to ensure genomic (and all) data

collected are shared according to both global and culturally-specific ethical standards, and considerations about confidentiality.

The following dataset was generated:

| Author(s) | Year | Dataset title | Dataset URL | Database and Identifier |
|-----------|------|---------------|-------------|-------------------------|
| Garcia AR | 2021 | APOE4 is associated with elevated blood lipids and lower levels of innate immune biomarkers in a tropical Amerindian subsistence population | https://doi.org/10.5061/dryad.pg4f4qrpt | Dryad Digital Repository, 10.5061/dryad.pg4f4qrpt |

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
