## [Decision Letter]

**Acceptance summary:**

The authors ask why the APOE4 allele has persisted, often at high frequencies, in human populations despite its associations to heart disease and Alzheimer's disease. They consider the hypothesis that APOE4 may be advantageous in a high pathogen and high physical environment settings (as opposed to a low pathogen industrial lifestyle) through an in-depth characterization of the Tsimane in Bolivia. The study is of broad interest with an insightful dataset; the scope of the project is clear, emphasizing the set of conclusions with appropriate support.

**Decision letter after peer review:**

Thank you for submitting your article "APOE4 is associated with elevated blood lipids and lower levels of innate immune biomarkers in a tropical Amerindian subsistence population" for consideration by *eLife*. Your article has been reviewed by 2 peer reviewers, and the evaluation has been overseen by a Reviewing Editor and George Perry as the Senior Editor. The reviewers have opted to remain anonymous.

The reviewers have discussed their reviews with one another, and the Reviewing Editor has drafted the following to help you prepare a revised submission.

Essential revisions:

1) Please provide more evolutionary biology context. For example, the paper would benefit from a quantitative discussion and context of APOE frequencies and pathogen prevalence in other worldwide populations. Are there other cross-population or within-population genomic signatures of selection in the region that would support the hypothesis of adaptation?

2) Relatedly, the null hypothesis for observed high frequency of the allele should be neutrality/drift until evidence mounts against it. This relates to both the structure/pitch of the study, as well as the statistical tests conducted. The number of statistical tests is high, and it is unclear if a multiple testing correction or other methods to prevent false positives were used. Was the study registered or preplanned tests, etc. Are the effect sizes biologically meaningful for the few tests that are significant?

3) Please clarify the reporting and/or interpretation of CRP values, and clarify other methodological issues throughout, such as specific sample sizes for each analysis and which alleles are considered, and units on figures. Importantly, please include underlying data points in the figures, not only plots of the estimated model.

*Reviewer #1 (Recommendations for the authors):*

1) Page 3, lines 114-123 and page 6, lines 234-237: Please see comment 1 in the above section. Something useful to compare allele frequencies could be the supplemental to Eisenberg et al. 2010, https://doi.org/10.1002/ajpa.21298 which calculated allele frequencies for various populations, or an aggregate database such as Ensemble could be used to aggregate the allele frequencies in various populations using the RS ids.

2) Page 6, lines 234-237 and page 16, lines 574-575: Please see comment 2 above. If other models weren't tested, was a power calculator used?

3) Briefly define the APOE3 and APOE4 alleles (CT vs CC) to the reader to make it accessible to readers unfamiliar with the nomenclature.

4) Page 5, Figure 1 is a little bit busy and hard to read, I would suggest a clean white background.

5) Page 14, lines 477-491. The population description paragraph is unnecessary in the Materials and methods given that this information is mostly repeated on page 5. Any information that is not present on the page 5 description, such as a "an upregulated immunity across the life course" should be added to it.

*Reviewer #2 (Recommendations for the authors):*

My thanks to the authors for putting together a nice manuscript. I thoroughly enjoyed reading it. It is well organized and interesting for a number of reasons. That said, I have several suggestions (no specific order below) that, I hope, will improve the final version of the paper.

– I believe that environmentally "moderated" rather than "mediated" should be used on Line 104. Please consider.

– The antagonistic pleiotropy hypothesis is well described and interesting. In my opinion, this should be more directly addressed in the Discussion. I am left wondering what more we know about population variation in APOE4 frequency (does it map onto variation in pathogenicity?) and why frequency is almost 4x higher in Central Africa than Tsimane. Drivers of frequency variation among subsistence populations (not just between subsistence and post-industrial) would be helpful to discuss.

– The choice of using "pathogen-diverse" instead of "high-pathogen" environments or similar throughout the paper should be explained. It is not clear to me what about diversity itself is thought to be important for this analysis.

– The sample sizes should be better clarified. In the Results text, it sounds like there are >6500 measures for each immune marker. There are, in fact, <1000 for CRP. Given the large variation in sample sizes for each measure and the fact that CRP and oxidized LDL are <1000 each, I suggest the authors find a way to include sample sizes in the main text, not just the SI. We are also told that all individuals included had at least 1 BMI and age measurement, but that is not my reading of Supp Table 1. It would be good to clarify. Please also include information about when the data were collected.

– I have several concerns and recommendations for the data treatment and analysis. Chiefly, I am concerned about the CRP data. The CRP values themselves that are reported (sample-wide mean listed as approx. 3.5 mg/dL) are not believable. This is likely an error in units reporting as the authors' 2016 paper lists mean Tsimane CRP around 1.5 mg/L ( = 0.15 mg/dL). It is hard to believe these values average 20+ times greater. Even if the actual mean CRP is 3.5 mg/L, this is still much higher than previously reported and requires explanation. The most likely explanation is that acute inflammatory events are likely often being captured as the sample has only <1.5x coverage of CRP measures. The work of McDade (2012) and others has shown that Amazonian subsistence populations have no evidence for chronic/persistent inflammation when measured repeatedly over short intervals. Such findings suggest that inflammation in pathogen-rich environments is not persistent but rather recurrent. This manuscript's argument is based on the CRP data reflecting "baseline" levels. I don't believe that this is the case. The inclusion of all CRP data in the analysis (even high values clearly indicative of active acute phase response) is problematic and makes the analysis of mean CRP levels very difficult to interpret. Are, for example, lower CRP values in APOE4E reflective of lower baseline levels, faster clearance following the resolution of acute CRP spike, or lower absolute magnitude of CRP spike? More information is needed on the WBC cut-off value used to determine current infection status. In general, given the nature of these data and their treatment, I am not convinced that reliable "baseline" measures of CRP are being used in the analysis.

– Some discussion of the limitations of using BMI as a proxy of energy availability is needed. This is a key measure in this analysis, but presumably BMI and fat mass have a variable relationship across the degree of market integration captured in this sample. Likewise, community as a proxy of pathogen exposure should be better justified. Were some participants from the same households? Is it helpful to also cluster by household in the analysis?

– Note that Figure 2, 4, etc. have no units listed for the y-axes.

[Editors' note: further revisions were suggested prior to acceptance, as described below.]

Thank you for submitting your article "APOE4 is associated with elevated blood lipids and lower levels of innate immune biomarkers in a tropical Amerindian subsistence population" for consideration by *eLife*. Your article has been reviewed by 2 peer reviewers, and the evaluation has been overseen by a Reviewing Editor and George Perry as the Senior Editor. The following individual involved in review of your submission has agreed to reveal their identity: Sam Urlacher (Reviewer #3).

The primary concerns from the original manuscript have been addressed, and a few smaller questions remain. Essential Revisions:

1) Consideration/discussion of pathogen diversity's relationship with inflammation and selection strength (discussed by reviewer 3).

2) Clarifying some statistical and sampling choices suggested by reviewers below.

*Reviewer #2 (Recommendations for the authors):*

In the previous submission, there were a few key elements the authors needed to address in a resubmission. The first was whether there was any evidence of selection for the APOE4 allele in the Tsimane population. The authors have reframed their approach to be less reliant on an argument for selection and more within an ecological context and how it relates to a metabolic mediated immune activation. APOE4 fits within that framework with respect to cholesterol and immunity and how it extends largely to a pathogen-driven environment. Furthermore, Dr. Garcia et al., have added a discussion on pages 10 (428-445), that seeks to describe contextually the broad mechanisms that could be responsible for the allele frequencies observed.

Another concern related to reporting the specifics from their statistical analyses as well as general concerns as to how the study was organized. To address this major concern, the authors have shifted some details earlier in the text, such as to how the genotypes were binned on page 7. This alleviated questions I had throughout reading the paper in the first submission. Furthermore, they have added more details about their statistical models and results into the text of the paper so that their discussion is much clearer.Their table now includes the number of observations used in the analyses. Lastly, they clarified the regression model and stated that they used FDR p-value adjustments to correct for multiple testing.

My only remaining question about their statistical analyses is whether they tested for any interactions between the variables in their regression models.

Overall, Dr. Garcia et al.,'s resubmission adequately answers my previous review concerns and I would recommend to acceptance to *eLife*.

---

## [Author Response]

Essential revisions:1) Please provide more evolutionary biology context. For example, the paper would benefit from a quantitative discussion and context of APOE frequencies and pathogen prevalence in other worldwide populations. Are there other cross-population or within-population genomic signatures of selection in the region that would support the hypothesis of adaptation?

Thank you for this input. We can understand why a reader may wonder if the frequency of APOE4 should be different in a population like the Tsimane where we lay out some conditions where the E4 allele may not have harmful effects on health. With regards to this point –raised also by Reviewer 1– it illuminates the fact that our paper would benefit from some clarification regarding the goals of the paper, and our position on APOE allele frequencies in the Tsimane population. Our goals are primarily focused on (1) questioning the assumptions that APOE4 is a universally deleterious allele, rather than its effects on phenotype being environmentally moderated, and (2) assessing the relationships between APOE, lipids, and innate inflammation in a population living in a relatively unique (and underrepresented) environmental context. It was not our intent to make a purely adaptationist argument for E4, or to suggest that there are no costs to E4 in other aspects of life, or at different ages, but rather to suggest that the relative costs and benefits of E4 may be environmentally-dependent, such that having an E4 allele is more neutral, and/or may provide some benefits (and limited costs), in pathogenically-diverse and energy-limited environments. Certainly some degree of genetic drift, antagonistic pleiotropy and other forces should be considered jointly to make inferences about the frequency of E4 in the Tsimane population. While pathogen risk and energetic limitations is one context where the typical costs of E4 may not be expressed, the high frequency of E4 allele in northern European populations confirms that more is involved in understanding population differences in E4 frequency. Nonetheless, the context we describe and show evidence for in our paper contrasts with the contemporary obesogenic environment with low pathogen diversity more typical of the Global North, where benefits like lipid buffering are no longer needed- and may in fact incur costs.

We have clarified and reframed the introduction to better couch the paper from an evolutionary biology perspective, and now discuss global variation in APOE allele frequencies, in ecological context, as well as clarifying our position on APOE in relation to adaptation. We include some excerpts below.

Examples of alterations to the Introduction:

"The evolutionary success of APOE3 has been attributed to its greater plasticity in response to environmental changes compared to the ancestral APOE4 allele (Huebbe and Rimbach, 2017). However, APOE4 is maintained in many human populations at nontrivial frequencies (as high as 40% in Central Africa) (Huebbe and Rimbach, 2017).

Maintenance and distribution of APOE variants may be in part due to the distinct functional capabilities of allelic variants (Trotter et al., 2011). Eisenberg and colleagues have suggested that the benefits of APOE4 may be most appreciable in environments where cholesterol requirements are increased to meet higher metabolic demands, such as at high- and low- latitudes where there are climatic extremes (e.g. northern Europe and South America) (Eisenberg et al., 2010)."

Paragraph added to the Discussion:

"Finally, nearly 1/4 of individuals in the sample had at least one copy of APOE4. The allelic frequency of APOE4 reported here is similar to what has been previously documented among other South American and Amerindian populations, which ranges from 5 – 30% and varies widely by region and level of genetic admixture (Corbo and Scacchp, 1999; Gayà-Vidal et al., 2012; Velez-Pardo et al., 2015). The distribution of APOE allelic variants in populations around the world is likely due to a mosaic of factors, potentially including differences related to environmentally-dependent costs and benefits of functionally-distinct variants. Certainly some degree of genetic drift or founder effects (Gayà-Vidal et al., 2012; Singh et al., 2006), antagonistic pleiotropy (Smith et al., 2019; Van Exel et al., 2017) and other forces (e.g. genetic relatedness) play an important role in determining population and global frequencies and should be considered jointly to make inferences about the frequency of E4 in the Tsimane population. While pathogen risk and energetic limitations is one context where the typical costs of E4 may not be expressed, the high frequency of E4 allele in populations that are appreciably different in terms of latitude, ecology, and population history (e.g. northern European and central African populations), confirm that more is involved in understanding population differences in E4 frequency (Abondio et al., 2019)."

Modifications to 1.2 Hypothesis and Aims:

"We hypothesize that, in pathogenically diverse and energy-limited contexts, an APOE4 variant is less harmful because it (a) minimizes damage caused by chronic innate inflammation, and (b) maintains higher circulating cholesterol levels, which buffer energetic costs of pathogen-driven innate immune activation."

2) Relatedly, the null hypothesis for observed high frequency of the allele should be neutrality/drift until evidence mounts against it. This relates to both the structure/pitch of the study, as well as the statistical tests conducted. The number of statistical tests is high, and it is unclear if a multiple testing correction or other methods to prevent false positives were used. Was the study registered or preplanned tests, etc. Are the effect sizes biologically meaningful for the few tests that are significant?

Our response to #1 (above) is also relevant to addressing the first point raised here. While we agree that the neutral/drift model should serve as the null hypothesis for allele frequency, it was not our intent to argue that E4 was at a particularly high frequency among the Tsimane, or test for signatures of selection in this paper. While these results could be used as preliminary rationale for those seeking to assess cross-population signatures of selection, we feel that focusing too much on this aspect would distract from the main goals of the paper. Specifically, this paper assesses the relationships between APOE, lipids, and innate inflammation. That being said, we do think it is an important point of discussion to address, and have added a paragraph to the Discussion section that speaks to global and more localized differences in APOE allelic frequencies (see above).

The high number of statistical tests is due to the broad aims of the paper, which are both descriptive and hypothesis-driven in nature. There are surprisingly few papers (particularly that have larger sample sizes) describing lipid and immune level variation by APOE genotype, and particularly poor coverage and representation of this type of data from non-US or European populations living in diverse ecologies. The inclusion of these descriptive comparisons will hopefully begin to fill a gap in the literature and contribute to understanding the basic science of how APOE interacts with lipids and immune function, as well as evaluating the role(s) of APOE in related diseases.

Regarding hypothesis-driven aims, we did conduct multiple testing correction for models that were included in testing our hypothesis related to the relationship between innate immune function and lipids under energetic conditions. However, while we do report FDR p-values in the main text, we realized that they were inadvertently left off from the tables. We now clarify what models are included in hypothesis-testing (see excerpt below), and revised tables accordingly, to add FDR p-values. To further improve clarity on this front, apart from Table 1 (which presents sample descriptives and test statistics from models comparing APOE3/3 vs. APOE4), we narrowed the tables and figures of the main text to present findings related to our hypothesis, and have moved the others to the Supplement. To address the final concern, we also now include more discussion on the biological meaningfulness of the tests that were significant:

In terms of effect sizes, we report on the biological relevance of CRP-differences (i.e. APOE4 carriers have 30% lower CRP compared to APOE33), and now extend this type of reporting to the other models.

The study was not pre-registered; however, the hypothesis was generated based on synthesizing previously published ideas (Trumble et al., 2017, Finch and Morgan 2007, Yassine and Finch 2020, Finch 2010, Trumble and Finch 2019, Eisenberg et al., 2010, Oria et al., 2005, van Exel et al., 2017). Specific biomarkers included in hypothesis-driven models were chosen either because their relationships with APOE were previously described in publications by co-authors, or based on extensive literature review on APOE and immune function.

Modification to 1.2 Hypothesis and Aims:

"This study describes the immunophenotypes and lipid profiles of individuals with APOE3/3 and APOE4+ genotypes living in a pathogenically-diverse, energy-limited environment. For the purpose of hypothesis-testing, we focus on testing genotype-related differences in components of innate immunity-- C-reactive protein (CRP), neutrophils, eosinophils, and erythrocyte sedimentation rate (ESR) lipids linked to inflammation (total cholesterol, LDL, and oxidized LDL). We evaluate the extent to which body mass index (BMI) moderates the association between lipids and innate inflammation (CRP, neutrophils, ESR). Finally, we test whether the APOE4 allele has a moderating effect on the relationship between BMI and blood lipids to evaluate the role of APOE4 in the maintenance of stable lipid levels under energetic restriction."

3) Please clarify the reporting and/or interpretation of CRP values, and clarify other methodological issues throughout, such as specific sample sizes for each analysis and which alleles are considered, and units on figures. Importantly, please include underlying data points in the figures, not only plots of the estimated model.

We had mistakenly reported CRP in mg/dL; CRP units have been updated and now correctly report in mg/L. See below (Reviewer 2) for a detailed response to our CRP reporting error. In terms of interpretation, while we did report on the biological relevance of CRP-differences (i.e. APOE4 carriers have 30% lower CRP compared to APOE33), we have extended this to other models.

Sample sizes, standardized betas, and standard errors, for main variables in each analysis have now been added to tables in the main text. Full models with covariates are still included in the Supplement. We have also updated all figures to include raw data points, and units on axes.

Reviewer #1 (Recommendations for the authors):1) Page 3, lines 114-123 and page 6, lines 234-237: Please see comment 1 in the above section. Something useful to compare allele frequencies could be the supplemental to Eisenberg et al. 2010, https://doi.org/10.1002/ajpa.21298 which calculated allele frequencies for various populations, or an aggregate database such as Ensemble could be used to aggregate the allele frequencies in various populations using the RS ids.

We believe that we have addressed this comment in our response to the comment 1 in Essential Revisions (above). Though an interesting and valuable question, we have opted not to explicitly compare allelic frequencies between the Tsimane and other populations, nor signatures of selection, but rather to evaluate the phenotypic associations of APOE variants and immune/lipid biomarkers just in this population. However, we have added more discussion of the global variance in APOE allelic frequencies to the Introduction and Discussion sections to provide important context (see above response to Essential Revisions for excerpts).

2) Page 6, lines 234-237 and page 16, lines 574-575: Please see comment 2 above. If other models weren't tested, was a power calculator used?

Similarly, we believe we have addressed this comment in response to comment 2 in Essential Revisions (above).

3) Briefly define the APOE3 and APOE4 alleles (CT vs CC) to the reader to make it accessible to readers unfamiliar with the nomenclature.

Sentences added to Section 1.1:

"APOE has three functionally polymorphic allelic variants: E4, E3, and E2 (Demarchi et al., 2005; Safieh et al., 2019). The most prevalent, APOE3, arose ~200K years ago from a single nucleotide polymorphism (C T) at locus 19q13 from the ancestral APOE4 (Huebbe and Rimbach, 2017)."

4) Page 5, Figure 1 is a little bit busy and hard to read, I would suggest a clean white background.

We lightened background and molecules substantially, and bolded words. However, we would like to include the background to aid in the visual distinction between the pre- and post- industrialized environments.

5) Page 14, lines 477-491. The population description paragraph is unnecessary in the Materials and methods given that this information is mostly repeated on page 5. Any information that is not present on the page 5 description, such as a "an upregulated immunity across the life course" should be added to it.

Thank you for pointing out the redundancy. We have integrated the non-redundant information, and have removed the redundant part from Materials and methods section.

Reviewer #2 (Recommendations for the authors):My thanks to the authors for putting together a nice manuscript. I thoroughly enjoyed reading it. It is well organized and interesting for a number of reasons. That said, I have several suggestions (no specific order below) that, I hope, will improve the final version of the paper.– I believe that environmentally "moderated" rather than "mediated" should be used on Line 104. Please consider.

We agree that 'moderated' is a more appropriate word choice. Changed.

– The antagonistic pleiotropy hypothesis is well described and interesting. In my opinion, this should be more directly addressed in the Discussion. I am left wondering what more we know about population variation in APOE4 frequency (does it map onto variation in pathogenicity?) and why frequency is almost 4x higher in Central Africa than Tsimane. Drivers of frequency variation among subsistence populations (not just between subsistence and post-industrial) would be helpful to discuss.

While we agree that making inferences about APOE4 frequency across populations is interesting and would be a useful extension, it is unfortunately beyond the scope of this manuscript. As we unpack above, the goal of this paper is to assess associations between APOE variants and immune and lipid physiology in a population facing relatively high pathogen stress. Given the important point raised regarding global variation, we do now briefly address these points in the Introduction and Discussion sections (see above in 'Essential Revisions' for excerpts).

– The choice of using "pathogen-diverse" instead of "high-pathogen" environments or similar throughout the paper should be explained. It is not clear to me what about diversity itself is thought to be important for this analysis.

Thank you for bringing this up. We do feel that the distinction is an important one, and is somewhat central to our hypothesis regarding how the costs and benefits of APOE are calibrated depending on environmental conditions. We have added a sentence to the introduction (section 1.1), and have aimed to better highlight this point throughout the paper. Just as a note, the importance of pathogen diversity in moderating the effects of APOE variants on metabolic and immune responses is from a theoretical standpoint, and at the population-level; we do not test the effects of different pathogens in this study.

Sentence added to Introduction:

"Maintaining lower levels of innate immunity may minimize the accumulative damage caused by low grade innate inflammation over the long term, while still enabling strong targeted immune responses to pathogens following exposure (Franceschi et al., 2000; Trumble and Finch, 2019), particularly in environments with a diversity of species of pathogens. For example, pathogens like helminthic parasites, help regulate and contain inflammation by triggering Th2-mediated (anti-inflammatory) immunological pathways (Maizels and McSorley, 2016; Motran et al., 2018), which may be important for counterbalancing a strong proinflammatory response."

– The sample sizes should be better clarified. In the Results text, it sounds like there are >6500 measures for each immune marker. There are, in fact, <1000 for CRP. Given the large variation in sample sizes for each measure and the fact that CRP and oxidized LDL are <1000 each, I suggest the authors find a way to include sample sizes in the main text, not just the SI. We are also told that all individuals included had at least 1 BMI and age measurement, but that is not my reading of Supp Table 1. It would be good to clarify. Please also include information about when the data were collected.

For clarity, we deleted the text at the start of the Results section (re: >6500 measures for each biomarker), and have added description of the data in the Methods sections (see excerpts below). We also opted to remove Supplemental Table 1, as we realized that it doesn't provide aptly describe the data as applied to models. To this end, all tables in the main text also now includes sample sizes, and we report sample sizes explicitly in the main text.

Text added regarding when data were collected:

"Biomarker data used in this paper were collected by the THLHP between 2004 – 2015 (see Gurven et al., 2017; Kraft et al., 2020 for details)."

Sentences added to explain differences in number of measures per individual for certain biomarkers:

"There are also fewer repeat measurements for a subset of biomarkers (i.e. C-reactive protein and oxidized LDL) that were assayed in the U.S., due to them being analyzed as part of a prior project. Specific sample sizes are reported in Table 1, and full tables report sample size for each model." (in Methods subsection: Sampling Design)

"Oxidized LDL and hs-CRP assays use materials from the same lot across all measures." (in Methods subsection: Measurement of blood lipids and immune function)

– I have several concerns and recommendations for the data treatment and analysis. Chiefly, I am concerned about the CRP data. The CRP values themselves that are reported (sample-wide mean listed as approx. 3.5 mg/dL) are not believable. This is likely an error in units reporting as the authors' 2016 paper lists mean Tsimane CRP around 1.5 mg/L ( = 0.15 mg/dL). It is hard to believe these values average 20+ times greater. Even if the actual mean CRP is 3.5 mg/L, this is still much higher than previously reported and requires explanation. The most likely explanation is that acute inflammatory events are likely often being captured as the sample has only <1.5x coverage of CRP measures. The work of McDade (2012) and others has shown that Amazonian subsistence populations have no evidence for chronic/persistent inflammation when measured repeatedly over short intervals. Such findings suggest that inflammation in pathogen-rich environments is not persistent but rather recurrent. This manuscript's argument is based on the CRP data reflecting "baseline" levels. I don't believe that this is the case. The inclusion of all CRP data in the analysis (even high values clearly indicative of active acute phase response) is problematic and makes the analysis of mean CRP levels very difficult to interpret. Are, for example, lower CRP values in APOE4E reflective of lower baseline levels, faster clearance following the resolution of acute CRP spike, or lower absolute magnitude of CRP spike?

Thank you so much for catching this reporting error: the CRP values are in mg/L. That the mean for this sample (3.5 mg/L) is somewhat higher than the means reported in Blackwell et al. 2016 is not unexpected. This is explained by the fact that CRP was measured only ~1x per individual, and the CRP measurements are mainly from when individuals were over the age of 40 years (the median age at time of CRP measurement for this sample is 54 years old – also see Supplemental Figure 1 which includes raw data points). As seen in Table 2 of Blackwell's 2016 paper, the mean CRP is 2.30mg/L in those 50+. While this is lower what we report, it is in sync, given that CRP increases across age, and our sample is biased to older individuals and extends to 93yrs old.

However, to address the possibility that we are capturing a high number of acute inflammatory events, which may affect findings, we reran models constraining them to include only observations of CRP < 10mg/L. The median level for CRP for this subset is 2.5mg/L. Constraining the models does not alter the results, however, we report the results for both, and include full output from the constrained model in the Supplement.

We see the problem of stating that the reported levels for CRP (or any biomarkers) are specifically "baseline", particularly given the observational nature of the data. A main focus of the paper is in applying an evolutionary lens to understanding relationships between APOE variants, lipids, and immune functions-- including the widely-observed phenomenon that in healthy, non-obese, individuals, APOE4 is consistently associated with lower CRP. We aim to apply an evolutionary theoretical framework to understand this relationship, however, with the existing data, we cannot make strong inferences or rule out the alternate explanations (lower baseline vs faster clearance, etc.) posed. As noted below, we have deleted all uses of the term "baseline" to describe observed levels of immune function. We have also added sentences to the Discussion section to illuminate the possibilities raised by this Reviewer.

From Discussion: "Our finding that innate immune biomarkers are lower among APOE4 carriers is in line with prior reports (Lumsden et al., 2020; Martiskainen et al., 2018; Trumble et al., 2017; Vasunilashorn et al., 2011), however the causes are uncertain. One proximate explanation involves the mevalonate pathway, which plays a key role in multiple cellular processes, including modulating sterol and cholesterol biosynthesis and innate immune function (Buhaescu and Izzedine, 2007). Regarding the main finding for CRP, it is possible that APOE4 carriers experience a lower innate immune sensing (Dose et al., 2018) or have faster clearance following the resolution of an acute spike. While there is currently no direct evidence for the latter, some studies have found that higher circulating lipids were associated with more rapid clearance of active infections (Andersen, 2018; Pérez-Guzmán et al., 2005)."

More information is needed on the WBC cut-off value used to determine current infection status.

Since there is not a "globally defined" standard for WBC cutoff to determine current infection, we referred to the range described in Williams and McKenzie's Clinical Laboratory Hematology text (2nd edition, 2010), which defines leukocytosis in adults as WBC > 11 x 10^9^/L to help determine the cutoff. However, we chose to use WBC > 12 x 10^9^/L as a cutoff, given the overall heightened immune function (without association with adverse health) among the Tsimane (Blackwell et al., 2016; Kaplan et al., 2017).

In addition to being described in the Methods section, we added a statement to the first paragraph of the Results section:

"Mixed effects multiple regression models were used to accommodate for multiple measurements per individual for some biomarkers, as well as community-level differences in pathogen exposures. Models also include covariates which adjust for age, sex, seasonality, and current infection (WBC > 12 x 10^9^/L) (see methods for details)."

In general, given the nature of these data and their treatment, I am not convinced that reliable "baseline" measures of CRP are being used in the analysis.

We appreciate this point, and see the problem of stating that the reported levels for CRP (or any biomarkers) are specifically "baseline", particularly given the observational nature of the data. We have deleted all uses of the term "baseline" to describe observed levels of immune function. Additionally, as mentioned above, we reran models constraining them to include only observations of CRP < 10mg/L (i.e. not indicative of acute infection). The median level for CRP for this subset is 2.5mg/L. Constraining the models does not alter the results, however, we report the results for both, and include constrained models in the Supplement.

– Some discussion of the limitations of using BMI as a proxy of energy availability is needed. This is a key measure in this analysis, but presumably BMI and fat mass have a variable relationship across the degree of market integration captured in this sample. Likewise, community as a proxy of pathogen exposure should be better justified. Were some participants from the same households? Is it helpful to also cluster by household in the analysis?

We agree that BMI is not a perfect proxy for energy availability and have added a sentence to the limitations section to acknowledge this point. However, given that our main goal in the paper -- with regards to energy availability -- was to investigate APOE and lipids at the extreme tails of BMI (overweight vs. lean), we do feel that BMI can adequately capture broad differences in energetic availability between these two groups. For example, a previous paper showed that BMI and body fat were closely associated among adults in this population (Gurven et al., 2012: r=.0.75 in women; r=0.57 in men; Figure S4). Also, though we use BMI as a continuous measure for models, we plot the upper and lower tertiles from these models to distinguish these overweight vs. lean groups.

Regarding justification for using community as a proxy for pathogen exposure, we have added the following sentence to Methods: "Because Tsimane villages vary in sanitation infrastructure, including access to soap and other hygienic products, and potentially prevalence by pathogen type (e.g. some living very close to the river versus farther out in the forest), individuals were clustered by community to account for variation in such community-level factors." We would also like to note that we include season and white blood cell count as additional covariates to adjust for differences in current pathogen exposure.

We appreciate the Reviewer's point regarding household-level differences in pathogen exposure, but given the age structure of the sample, the maximum number of individuals that could be in the same house would be two (married couple). If we had multiple children per household in the sample, then we would have controlled for household, but given the age structure of the sample, community of residence and season provide a greater capture of pathogen variation than household.

Sentences added to Limitations:

"Third, proxies for energy availability and pathogen exposures are imperfect. BMI and fat free mass may have a variable relationship across the degree of market integration. However, given that our main goal in the paper – with regards to energy availability – is to investigate APOE and lipids at the extreme tails of BMI (lean vs. obese), BMI should adequately capture broad differences in energetic availability between these two groups."

– Note that Figure 2, 4, etc. have no units listed for the y-axes.

Thank you for noting this. We have made sure that all Figures now include units on all axes.

[Editors' note: further revisions were suggested prior to acceptance, as described below.]

The reviewers have discussed their reviews with one another, and the Reviewing Editor has drafted this to help you prepare a revised submission.The primary concerns from the original manuscript have been addressed, and a few smaller questions remain. Essential revisions:1) Consideration/discussion of pathogen diversity's relationship with inflammation and selection strength (discussed by reviewer 3).2) Clarifying some statistical and sampling choices suggested by reviewers below.

First, we express gratitude to the reviewers for their comments and helpful suggestions throughout the review process, and thank them and the editorial board for the opportunity to have our manuscript considered for publication in eLife. Below we address the remaining questions and comments from each reviewer.

Reviewer #2 (Recommendations for the authors):In the previous submission, there were a few key elements the authors needed to address in a resubmission. The first was whether there was any evidence of selection for the APOE4 allele in the Tsimane population. The authors have reframed their approach to be less reliant on an argument for selection and more within an ecological context and how it relates to a metabolic mediated immune activation. APOE4 fits within that framework with respect to cholesterol and immunity and how it extends largely to a pathogen-driven environment. Furthermore, Dr. Garcia et al. have added a discussion on pages 10 (428-445), that seeks to describe contextually the broad mechanisms that could be responsible for the allele frequencies observed.Another concern related to reporting the specifics from their statistical analyses as well as general concerns as to how the study was organized. To address this major concern, the authors have shifted some details earlier in the text, such as to how the genotypes were binned on page 7. This alleviated questions I had throughout reading the paper in the first submission. Furthermore, they have added more details about their statistical models and results into the text of the paper so that their discussion is much clearer.Their table now includes the number of observations used in the analyses. Lastly, they clarified the regression model and stated that they used FDR p-value adjustments to correct for multiple testing.My only remaining question about their statistical analyses is whether they tested for any interactions between the variables in their regression models.

We did not systematically test for all possible interactions between APOE and covariates (or covariates themselves) in analyses for this paper. We did, however, test for collinearity between variables included in the same model by assessing variance inflation factors (VIF) and correlations between fixed effects. No models had a VIF >2, and no variables had a correlation > ±0.5.

That being said, interactive effects between APOE and other phenotypic features (e.g. sex, age) are definitely of interest. There are several reasons we did not test for additional interactions here. First, to comprehensively interrogate interactions would require more space and a more targeted analytical approach, which we leave for the focus of a future analysis. Second, we lack a strong theoretical hypothesis at the present for a particular interactive effect, and we would like to avoid a fishing expedition with our observational data. If there is a specific, theoretically motivated interaction effect that the reviewer would like to see included, we would welcome the suggestion for inclusion.